# On $\mathcal{O}(1/K)$ Convergence and Low Sample Complexity for Single-Timescale Policy Evaluation with Nonlinear Function Approximation

## Abstract

Learning an accurate value function for a given policy is a critical step in solving reinforcement learning (RL) problems. So far, however, the convergence speed and sample complexity performances of most existing policy evaluation algorithms remain unsatisfactory, particularly with *non-linear* function approximation. This challenge motivates us to develop a new variance-reduced primal-dual method (VRPD) that is able to achieve a fast convergence speed for RL policy evaluation with nonlinear function approximation. To lower the high sample complexity limitation of variance-reduced approaches (due to the periodic full gradient evaluation with all training data), we further propose an enhanced VRPD method with an adaptive-batch adjustment (VRPD$^+$). The main features of VRPD include: i) VRPD allows the use of *constant* step sizes and achieves the $\mathcal{O}(1/K)$ convergence rate to the first-order stationary points of non-convex policy evaluation problems; ii) VRPD is a generic *single*-timescale algorithm that is also applicable for solving a large class of non-convex strongly-concave minimax optimization problems; iii) By adaptively adjusting the batch size via historical stochastic gradient information, VRPD$^+$ is more sample-efficient empirically without loss of theoretical convergence rate. Our extensive numerical experiments verify our theoretical findings and showcase the high efficiency of the proposed VRPD and VRPD$^+$ algorithms compared with the state-of-the-art methods.

## 1 Introduction

In recent years, advances in reinforcement learning (RL) have achieved enormous successes in a large number of areas, including healthcare (Petersen et al., 2019; Raghu et al., 2017b), financial recommendation (Theocharous et al., 2015), resources management (Mao et al., 2016; Tesauro et al., 2006) and robotics (Kober et al., 2013; Levine et al., 2016; Raghu et al., 2017a), to name just a few. In RL applications, an agent interacts with an environment and repeats the tasks of observing the current state, performing a policy-based action, receiving a reward, and transition to the next state. A key step in many RL algorithms is the policy evaluation (PE) problem, which aims to learn the value function that estimates the expected long-term accumulative reward for a given policy. Value functions not only explicitly provide the agent's accumulative rewards, but could also be utilized to update the current policy so that the agent can visit valuable states more frequently (Bertsekas & Tsitsiklis, 1995; Lagoudakis & Parr, 2003). In RL policy evaluation, two of the most important performance metrics are *convergence rate* and *sample complexity*. First, since policy evaluation is a subroutine of an overall RL task, developing fast-converging policy evaluation algorithms is of critical importance to the overall efficiency of RL. Second, due to the challenges in collecting a large number of training samples (trajectories of state-action pairs) for policy evaluations in RL, reducing the number of samples (i.e., sample complexity) can significantly alleviate the burden of data collection for solving policy evaluation problems. These two important aspects motivate us to pursue a fast-converging policy evaluation algorithm with a low sample-complexity in this paper.

Among various algorithms for policy evaluation, one of the simplest and most effective methods is the temporal difference (TD) learning approach (Sutton, 1988). Instead of focusing on the predicted and actual outcomes, the key idea of the TD learning is to make the difference between temporally successive predictions small. Specifically, the TD learning approach learns the value function by

using the Bellman equation to bootstrap from the current estimated value function. To date, there have been many algorithms proposed within the family of TD learning (Dann et al., 2014). However, most of these methods suffer from either a unstable convergence performance, (e.g., TD($\lambda$) (Sutton, 1988) for off-policy training) or a high computational complexity (e.g., least-squares temporal difference (LSTD) (Boyan, 2002; Bradtke & Barto, 1996)) in training with massive features. The limitation of these early attempts is largely due to the fact that they do not leverage the gradient-oracle in policy evaluation. Thus, in recent years, gradient-based policy evaluation algorithms have become increasingly prevalent. However, the design of efficient gradient-based policy evaluation algorithm is a non-trivial task. On one hand, as an RL task becomes more sophisticated, it is more appropriate to utilize *nonlinear function approximation* (e.g., deep neural network (DNN)) to model the value function. However, when working with nonlinear DNN models, the convergence performance of the conventional single-timescale TD algorithms may not be guaranteed (Tsitsiklis & Van Roy, 1997). To address this issue, some convergent two-timescale algorithms (Bhatnagar et al., 2009; Chung et al., 2018) have been proposed at the expense of higher implementation complexity. On the other hand, modern policy evaluation tasks could involve a large amount of state transition data. To perform policy evaluation, algorithms typically need to calculate *full gradients* that require all training data (e.g., gradient temporal difference (GTD) (Sutton et al., 2008) and TD with gradient correction (TDC) (Sutton et al., 2009b)), which entails a high sample complexity. So far, existing works on PE are either focus on linear approximation (GTD2 (Sutton et al., 2009b), PDBG (Du et al., 2017), SVRG (Du et al., 2017), SAGA (Du et al., 2017)) or have such a slower convergence performance (STSG (Qiu et al., 2020), VR-STSG (Qiu et al., 2020), nPD-VR (Wai et al., 2019)) (see detailed discussions in Section. 2). In light of the above limitations, in this paper, we ask the following question: *Could we develop an efficient single-timescale gradient-based algorithm for policy evaluation based on nonlinear function approximation?*

In this paper, we give an *affirmative* answer to the above question. Specifically, we propose an efficient gradient-based variance-reduced primal-dual algorithm (VRPD) to tackle the policy evaluation problem with nonlinear function approximation, which we recast as a minimax optimization problem. Our VRPD algorithm admits a simple and elegant *single-timescale* algorithmic structure. Then, we further enhance VRPD by proposing VRPD$^+$, which uses adaptive batch sizes to relax the periodic full gradient evaluation to further reduce sample complexity. The main contribution of this paper is that our proposed algorithms achieve an $\mathcal{O}(1/K)$ convergence rate ($K$ is the number of iterations) with *constant step-sizes* for policy evaluation with *nonlinear* function approximation, which is the *best-known* result in the literature thus far. Our main results are highlighted as follows:

- By utilizing a variance reduction technique, our VRPD algorithm allows *constant step-sizes* and enjoys a low sample complexity. We show that, under mild assumptions and appropriate parameter choices, VRPD achieves an $\mathcal{O}(1/K)$ convergence rate to the first-order stationary point of a class of nonconvex-strongly-concave(NCSC) minimax problems, which is the best-known result in the literature. To achieve this result, our convergence rate analysis introduces new proof techniques and resolves an open question and clarifies an ambiguity in the state-of-the-art convergence analysis of VR-based policy evaluation methods (see 2nd paragraph in Section 2.1 for more discussions).

- VRPD$^+$ significantly improves the sample complexity of the VRPD algorithm for policy evaluation with massive datasets. Our VRPD$^+$ (adaptive-batch VRPD) algorithm incorporates historical information along the optimization path, but does not involve backtracking and condition verification. We show that our VRPD$^+$ algorithm significantly reduces the number of samples and the computation loads of gradients, thanks to our proposed adaptive batch size technique that is able to avoid full gradient evaluation.

- Our extensive experimental results also confirm that our algorithms outperform the state-of-the-art gradient-based policy evaluation algorithms, and our VRPD$^+$ can further reduce the sample complexity compared to the VRPD algorithm. It is worth noting that, although the focus of our work is on RL policy evaluation, our algorithmic design and proof techniques contribute to the area of minimax optimization and could be of independent theoretical interest.

## 2 RELATED WORK

**1) TD Learning with Function Approximation for Policy Evaluation:** TD learning with function approximation plays a vital role in policy evaluation for RL. The key idea of TD learning is to

Table 1: Algorithms comparison for solving policy evaluation. $M$ is the size of the dataset; $K$ is the total iteration.

| Algorithm | Function Approx. | Problem | Step-size | Convergence Rate |
|---|---|---|---|---|
| GTD2 Sutton et al. (2009b) | Linear | - | $\mathcal{O}(1)$ | - |
| PDBG Du et al. (2017) | Linear | Convex-Concave | $\mathcal{O}(1)$ | $\mathcal{O}(1/K)$ |
| SVRG Du et al. (2017) | Linear | Convex-Concave | $\mathcal{O}(1)$ | $\mathcal{O}(1/K)$ |
| SAGA Du et al. (2017) | Linear | Convex-Concave | $\mathcal{O}(1)$ | $\mathcal{O}(1/K)$ |
| STSG Qiu et al. (2020) | Nonlinear | Stochastic/NCSC | $\mathcal{O}(1)$ | $\mathcal{O}(1/K^{1/2})$ |
| VR-STSG Qiu et al. (2020) | Nonlinear | Stochastic/ NCSC | $\mathcal{O}(1)$ | $\mathcal{O}(1/K^{2/3})$ |
| nPD-VR Wai et al. (2019) | Nonlinear | Finte-Sum / NCSC | $\mathcal{O}(1/M)$ | Conjectured[1] $\mathcal{O}(1/K)$ |
| **VRPD [Ours.]** | Nonlinear | Finte-Sum / NCSC | $\mathcal{O}(1)$ | $\mathcal{O}(1/K)$ |
| **VRPD$^{+}$ [Ours.]** | Nonlinear | Finte-Sum / NCSC | $\mathcal{O}(1)$ | $\mathcal{O}(1/K)$ |

[1] Whether nPD-VR achieves $\mathcal{O}(1/K)$ convergence rate remains an open question. See the detailed discussions in Sections 2 and 4.

minimize the Bellman error for approximating the value function. So far, most existing TD learning algorithms with theoretical guarantees focus on *the linear setting* (e.g., (Sutton et al., 2009a; Srikant & Ying, 2019; Xu et al., 2020b; Stankovic & Stankovic, 2016; Touati et al., 2018)). Doan et al. (2019), Liu et al. (2015), Macua et al. (2014), and Zhang & Xiao (2019) provided a finite-time analysis for the proposed distributed TD(0) and showed that the convergence rate of their algorithm is $\mathcal{O}(1/K)$. It was shown in Du et al. (2017) that policy evaluation with linear function approximation by TD(0) can be formulated as a strongly convex-concave or convex-concave problem, and can be solved by a primal-dual method with a linear convergence rate. However, the linearity assumption cannot be applied in a wide range of policy evaluations with nonlinear models. TD learning with nonlinear (smooth) function approximation is far more complex. Maei et al. (2009) was among the first to propose a general framework for minimizing the generalized mean-squared projected Bellman error (MSPBE) with smooth and *nonlinear* value functions. However, they adopted *two-timescale* step-sizes but only obtained a slow convergence performance. Other TD methods with nonlinear function approximations for policy evaluations include (Wang et al., 2017; 2016). Qiu et al. (2020) also investigated nonlinear TD learning and proposed two single-timescale first-order stochastic algorithms. However, the convergence rate of their STSG and VR-STSG are $\mathcal{O}(1/K^{1/4})$ and $\mathcal{O}(1/K^{1/3})$, while our VRPD algorithm achieves a *much faster* $\mathcal{O}(1/K)$ convergence rate.

In policy evaluation with non-linear function approximation, the state-of-the-art and the most related work to ours is (Wai et al., 2019), which showed that minimizing the generalized MSPBE problem is equivalent to solving a non-convex-strongly-concave (NCSC) minimax optimization problem via the Fenchel's duality. However, their best convergence results only hold when the step-size is $\mathcal{O}(\frac{1}{M})$, where $M$ is the size of the dataset. This is problematic for modern RL problems with a large state-action transition dataset. More importantly, although their convergence theorem appears to have a $\frac{1}{K}$ factor ($K$ being the total number of iterations), their convergence rate bound is in the form of $\frac{F^{(K)}+\text{Constant1}}{K\cdot\text{Constant2}}$ (cf. Theorem 1, Eq. (26) in Wai et al. (2019)). Notably, the $F^{(K)}$ term in the denominator in Eq. (26) inherently depends on the primal and dual values $\boldsymbol{\theta}^{(K)}$ and $\boldsymbol{\omega}^{(K)}$ in the $K$-th iteration, respectively. It is unclear whether $\boldsymbol{\omega}^{(K)}$ can be bounded in (Wai et al., 2019), hence leading to an ambiguity in guaranteeing an $\mathcal{O}(1/K)$ convergence rate. Thus, *whether an $\mathcal{O}(1/K)$ convergence rate is achievable in single-timescale policy evaluation with nonlinear function approximation and constant step-sizes remains an open question thus far.* **The key contribution and novelty** in this paper is that we resolve the above open question by proposing two new algorithms, both achieving an $\mathcal{O}(1/K)$ convergence rate. To establish this result, we propose a **new convergence metric** (cf. Eq. (9) in Section 4.1), which necessitates new proof techniques and analysis. For easy comparisons, we summarize our algorithms and the related works in Table 1.

**2) Relations with NCSC Minimax Optimization:** Although the focus of our paper is on RL policy evaluation, our algorithmic techniques are also related to the area of NCSC minimax optimization due to the primal-dual MSPBE formulation (cf. Eq. (2) in Section 3). Early attempts in (Nouiehed et al., 2019; Lin et al., 2020b) developed gradient descent-ascent algorithms to solve the NCSC minimax problems. However, these methods suffer from a high sample complexity and slow convergence rate. To overcome this limitation, two variance-reduction algorithms named SREDA (Luo et al., 2020) are proposed for solving NCSC minimax problems, which shares some similarity to our work.

Later, Xu et al. (2020a) enhanced SREDA to allow bigger step-sizes. However, our algorithms still differ from SREDA in the following key aspects: (i) Our algorithms are *single-timescale* algorithms, which are much easier to implement. In comparison, SREDA is a two-timescale algorithm, where solving an inner concave maximization subproblem is needed. Thus, to a certain extent, SREDA can be viewed as a triple-loop structure, and hence the computational complexity of SREDA is higher than ours; (ii) In the initialization stage, SREDA uses the PiSARAH, which is a subroutine that aims to help the SREDA algorithm achieve the desired accuracy at the initialization step and can be seen as an additional step to solve an inner concave maximization subproblem. Thus, SREDA has a higher computation cost than our paper. (iii) The number of parameters in SREDA are far more than ours and it requires the knowledge of the condition number to set the algorithm's parameters for good convergence performance. By contrast, our algorithms only require step-sizes $\alpha$ and $\beta$ to be sufficiently small, which is easier to tune in practice. (iv) SREDA does *not* provide an explicit convergence rate in their paper (it is unclear what their convergence rate is from their proof either). Yet, we show that our VRPD in theory has a lower sample complexity than that of SREDA.

Another related work in terms of NCSC minimax optimization is (Zhang et al., 2021), which also provided sample complexity upper and lower bounds. However, there remains a gap between the sample complexity lower and upper bounds in (Zhang et al., 2021). By contrast, the sample complexity of our VRPD algorithm *matches the lower bound* $\mathcal{O}(M + \sqrt{M}\epsilon^{-2})$ in (Zhang et al., 2021), which is the *first* in the literature. Furthermore, the algorithm contains an inner minimax subproblem (cf. Line 6 of Algorithm 1 in Zhang et al. (2021)). Solving such a subproblems in the inner loop incurs high computational costs. Due to this reason, the algorithm in (Zhang et al., 2021) had to settle for an inexact solution, which hurts the convergence performance in practice. In contrast, our algorithm does not have such a limitation.

## 3 PRELIMINARIES AND PROBLEM STATEMENT

We start from introducing the necessary background of reinforcement learning, with a focus on the policy evaluation problem based on nonlinear function approximation.

**1) Policy Evaluation with Nonlinear Approximation:** RL problems are formulated using the Markov decision process (MDP) framework defined by a five-tuple $\{\mathcal{S}, \mathcal{A}, P, \gamma, \mathcal{R}\}$, where $\mathcal{S}$ denotes the state space and $\mathcal{A}$ is the action space; $P : \mathcal{S} \times \mathcal{A} \to \mathcal{S}$ represents the transition function, which specifies the probability of one state transitioning to another after taking an action; $\mathcal{R}$ denotes the space of the received reward upon taking an action $a \in \mathcal{A}$ under state $s \in \mathcal{S}$ (in this paper, we assume that the state and action spaces are finite, but the numbers of states and actions could be large); and $\gamma \in [0, 1)$ is a time-discount factor.

For RL problems over an infinite discrete-time horizon $\{t \in \mathbb{N}\}$, the learning agent executes an action $a_t$ according to the state $s_t$ and some policy $\pi : \mathcal{S} \to \mathcal{A}$. The system then transitions into a new random state $s_{t+1}$ in the next time-slot. Also, the agent receives a random reward $R^\pi(s_t, a_t)$. The trajectory generated by a policy $\pi$ is a sequence of state-action pairs denoted as $\{s_1, a_1, s_2, a_2, \ldots\}$. The goal of the agent is to learn an optimal policy $\pi^*$ to maximize the long-term discounted total reward. Specifically, for a policy $\pi$ (could be a randomized policy), the expected reward received by the agent at state $s$ in any given time-slot can be computed as $R^\pi(s_t) = \mathbb{E}_{a \sim \pi(\cdot|s)}[R^\pi(s_t, a)]$. The value function $V^\pi(s_0) = \mathbb{E}[\sum_{t=0}^\infty \gamma^t R(s_t) \mid s_0, \pi]$ indicates the long-term discounted reward of policy $\pi$ over an infinite horizon with the initial state at $s_0 \in \mathcal{S}$. Also, the Bellman equation implies that $V^\pi(\cdot)$: $V(s) = \mathcal{T}^\pi V(s)$, where $\mathcal{T}^\pi f(s) \triangleq \mathbb{E}[R^\pi(s) + \gamma f(s')|a \sim \pi(\cdot|s), s' \sim P(\cdot|s, a)]$ denotes the Bellman operator. In RL, the agent's goal is to determine an optimal policy $\pi^*$ that maximizes the value function $V^\pi(s)$ from any initial state $s$.

However, the first obstacle in solving RL problems stems from evaluating $V^\pi(\cdot)$ for a given $\pi$ since $P(\cdot|s, a)$ is unknown. Moreover, it is often infeasible to store $V^\pi(s)$ since the state space $\mathcal{S}$ could be large. To address these challenges, one popular approach in RL is to approximate $V^\pi(\cdot)$ using a family of parametric and smooth functions in the form of $V^\pi(\cdot) \approx V_{\boldsymbol{\theta}^\pi}(\cdot)$, where $\boldsymbol{\theta}^\pi \in \mathbb{R}^d$ is a $d$-dimensional parameter vector. Here, $\Theta$ is a compact subspace. For notational simplicity, we will omit all superscripts "$\pi$" whenever the policy $\pi$ is clear from the context. In this paper, we focus on *nonlinear* function approximation, i.e., $V_{\boldsymbol{\theta}}(\cdot) : \mathcal{S} \to \mathbb{R}$ is a nonlinear function with respect to (w.r.t.) $\boldsymbol{\theta}$. For example, $V_{\boldsymbol{\theta}}(\cdot)$ could be based on a $\boldsymbol{\theta}$-parameterized nonlinear DNN. We assume that the gradient and Hessian of $V_{\boldsymbol{\theta}}(\cdot)$ exist and are denoted as: $g_{\boldsymbol{\theta}}(s) := \nabla_{\boldsymbol{\theta}} V_{\boldsymbol{\theta}}(s) \in \mathbb{R}^d, H_{\boldsymbol{\theta}}(s) := \nabla_{\boldsymbol{\theta}}^2 V_{\boldsymbol{\theta}}(s) \in \mathbb{R}^{d \times d}$.

Our goal is to find the optimal parameter $\boldsymbol{\theta}^* \in \mathbb{R}^d$ that minimizes the error between $V_{\boldsymbol{\theta}^*}(\cdot)$ and $V(\cdot)$. This problem can be formulated as minimizing the mean-squared projected Bellman error (MSPBE) of the value function as follows (Liu et al., 2018):

$$\mathrm{MSPBE}(\boldsymbol{\theta}) := \frac{1}{2} \big\| \mathbb{E}_{\mathbf{s} \sim D^\pi(\cdot)} \big[ (\mathcal{T}^\pi V_{\boldsymbol{\theta}}(s) - V_{\boldsymbol{\theta}}(s)) \nabla_{\boldsymbol{\theta}} V_{\boldsymbol{\theta}}(s)^\top \big] \big\|_{\boldsymbol{D}^{-1}}^2$$

$$= \max_{\boldsymbol{\omega} \in \mathbb{R}^d} \big( -\frac{1}{2} \mathbb{E}_{\mathbf{s} \sim D^\pi(\cdot)} [(\boldsymbol{\omega}^\top g_\theta(s))^2] + \langle \boldsymbol{\omega}, \mathbb{E}_{\mathbf{s} \sim D^\pi(\cdot)} [(\mathcal{T}^\pi V_{\boldsymbol{\theta}}(s) - V_{\boldsymbol{\theta}}(s)) g_\theta(s)] \rangle \big), \qquad (1)$$

where $D^\pi(\cdot)$ is the stationary distribution of under policy $\pi$ and $\boldsymbol{D} = \mathbb{E}_{\mathbf{s} \sim D^\pi}[g_\theta(s) g_\theta^\top(s)] \in \mathbb{R}^{d \times d}$.

**2) Primal-Dual Optimization for MSPBE:** It is shown in (Liu et al., 2018) (cf. Proposition 1) that minimizing MSPBE$(\boldsymbol{\theta})$ in (1) is equivalent to solving a primal-dual minimax optimization problem:

$$\min_{\boldsymbol{\theta} \in \mathbb{R}^d} \max_{\boldsymbol{\omega} \in \mathbb{R}^d} L(\boldsymbol{\theta}, \boldsymbol{\omega}), \qquad (2)$$

where $L(\boldsymbol{\theta}, \boldsymbol{\omega}) \triangleq \langle \boldsymbol{\omega}, \mathbb{E}_{\mathbf{s} \sim D^\pi(\cdot)} [(\mathcal{T}^\pi V_{\boldsymbol{\theta}}(s) - V_{\boldsymbol{\theta}}(s)) g_\theta(s)^\top ] \rangle - \frac{1}{2} \mathbb{E}_{\mathbf{s} \sim D^\pi(\cdot)} [(\boldsymbol{\omega}^\top g_\theta(s))^2]$. Since the distribution $D^\pi(\cdot)$ is unknown and the expectation cannot be evaluated directly, one often considers the following empirical minimax problem by replacing the expectation in $L(\boldsymbol{\theta}, \boldsymbol{\omega})$ with a finite sample average approximation in the stochastic objective function based on an $M$-step trajectory, i.e., $\min_{\boldsymbol{\theta} \in \mathbb{R}^d} \max_{\boldsymbol{\omega} \in \mathbb{R}^d} \mathcal{L}(\boldsymbol{\theta}, \boldsymbol{\omega}) = \min_{\boldsymbol{\theta} \in \mathbb{R}^d} \max_{\boldsymbol{\omega} \in \mathbb{R}^d} \frac{1}{M} \sum_{i=1}^M \mathcal{L}_i(\boldsymbol{\theta}, \boldsymbol{\omega})$, where

$$\mathcal{L}_i(\boldsymbol{\theta}, \boldsymbol{\omega}) := \langle \boldsymbol{\omega}, [R(s_i, a_i, s_{i+1}) + \gamma V_{\boldsymbol{\theta}}(s_{i+1}) - V_{\boldsymbol{\theta}}(s_i)] \times g_\theta(s_i) \rangle - \frac{1}{2} (\boldsymbol{\omega}^\top g_\theta(s_i))^2. \qquad (3)$$

Solving the above empirical minimax problem for MSPBE constitutes the rest of this paper.

## 4 SOLUTION APPROACH

As mentioned in Section 3, based on an $M$-step trajectory $\{s_1, a_1 \cdots, s_M, a_M, s_{M+1}\}$ generated by some policy $\pi$, our goal is to solve the empirical primal-dual and finite-sum optimization problem:

$$\min_{\boldsymbol{\theta} \in \mathbb{R}^d} \max_{\boldsymbol{\omega} \in \mathcal{W}} \frac{1}{M} \sum_{i=1}^M \mathcal{L}_i(\boldsymbol{\theta}, \boldsymbol{\omega}) = \min_{\boldsymbol{\theta} \in \mathbb{R}^d} \max_{\boldsymbol{\omega} \in \mathcal{W}} \mathcal{L}(\boldsymbol{\theta}, \boldsymbol{\omega}), \qquad (4)$$

where $\mathcal{W}$ is assumed to be a convex constrained set (Problem (4) becomes Problem (2) when $\mathcal{W} = \mathbb{R}^d$). In our Appendix, we also discussed the min-max problem while $\theta \in \Theta$. $\Theta$ is a convex constrained set. See details in Appendix. 12. Note that Problem (4) could be non-convex (e.g., DNN-based nonlinear approximation). Let $J(\boldsymbol{\theta}) \triangleq \max_{\boldsymbol{\omega} \in \mathcal{W}} \mathcal{L}(\boldsymbol{\theta}, \boldsymbol{\omega})$. Then, we can equivalently rewrite Problem (4) as follows: $\min_{\boldsymbol{\theta} \in \mathbb{R}^d} \max_{\boldsymbol{\omega} \in \mathcal{W}} \mathcal{L}(\boldsymbol{\theta}, \boldsymbol{\omega}) = \min_{\boldsymbol{\theta} \in \mathbb{R}^d} J(\boldsymbol{\theta})$.

Note from (3) that $\mathcal{L}(\boldsymbol{\theta}, \boldsymbol{\omega})$ is strongly concave w.r.t. $\boldsymbol{\omega}$, which guarantees the existence and uniqueness of the solution to the problem $\max_{\boldsymbol{\omega} \in \mathcal{W}} \mathcal{L}(\boldsymbol{\theta}, \boldsymbol{\omega}), \forall \boldsymbol{\theta} \in \mathbb{R}^d$. Then, given $\boldsymbol{\theta} \in \mathbb{R}^d$, we define the following notation: $\boldsymbol{\omega}^*(\boldsymbol{\theta}) := \operatorname*{argmax}_{\boldsymbol{\omega} \in \mathcal{W}} \mathcal{L}(\boldsymbol{\theta}, \boldsymbol{\omega})$. Thus, $J(\boldsymbol{\theta})$ can be further written as:

$$J(\boldsymbol{\theta}) = \mathcal{L}(\boldsymbol{\theta}, \boldsymbol{\omega}^*) = \max_{\boldsymbol{\omega} \in \mathcal{W}} \mathcal{L}(\boldsymbol{\theta}, \boldsymbol{\omega}). \qquad (5)$$

The function $J(\boldsymbol{\theta})$ can be viewed as a finite empirical version of MSPBE. We aim to minimize $J(\boldsymbol{\theta})$ by finding the stationary point of $\mathcal{L}(\boldsymbol{\theta}, \boldsymbol{\omega})$. To simplify the notaion, we use $\boldsymbol{\omega}^*$ to denote $\boldsymbol{\omega}^*(\boldsymbol{\theta})$. Note that if $\boldsymbol{D}$ in Eq. (1) is positive definite, Problem (4) is strongly concave in $\boldsymbol{\omega}$, but non-convex in $\boldsymbol{\theta}$ in general due to the non-convexity of function $V_{\boldsymbol{\theta}}$. Thus, the stated primal-dual objective function is a NCSC optimization problem. In this paper, we make the following assumptions:

*Assumption* 1 ($\mu$-Strongly Concavity). The differentiable function $\mathcal{L}(\boldsymbol{\theta}, \boldsymbol{\omega})$ is $\mu$-strongly concave in $\boldsymbol{\omega}$: if $\mathcal{L}(\boldsymbol{\theta}, \boldsymbol{\omega}) \leq \mathcal{L}(\boldsymbol{\theta}, \boldsymbol{\omega}') + \nabla_{\boldsymbol{\omega}} \mathcal{L}(\boldsymbol{\theta}, \boldsymbol{\omega}')^\top (\boldsymbol{\omega} - \boldsymbol{\omega}') - \frac{\mu}{2} \|\boldsymbol{\omega} - \boldsymbol{\omega}'\|^2, \forall \boldsymbol{\omega}, \boldsymbol{\omega}' \in \mathbb{R}^d, \mu > 0$ and any fixed $\boldsymbol{\theta} \in \mathbb{R}^d$. The above mentioned condition is equivalent to : $\|\nabla_{\boldsymbol{\omega}} \mathcal{L}(\boldsymbol{\theta}, \boldsymbol{\omega}) - \nabla_{\boldsymbol{\omega}} \mathcal{L}(\boldsymbol{\theta}, \boldsymbol{\omega}')\| \geq \mu \|\boldsymbol{\omega} - \boldsymbol{\omega}'\|, \forall \boldsymbol{\omega}, \boldsymbol{\omega}' \in \mathbb{R}^d$. Similar proofs can be found in Lemma 2 and 3 in Zhou (2018).

*Assumption* 2 ($L_f$-Smoothness). For $i = 1, 2, \ldots, M$, both gradient $\nabla_{\boldsymbol{\theta}} \mathcal{L}_i(\boldsymbol{\theta}, \boldsymbol{\omega})$ and $\nabla_{\boldsymbol{\omega}} \mathcal{L}_i(\boldsymbol{\theta}, \boldsymbol{\omega})$ are $L_f$-smooth. That is, for all $\boldsymbol{\theta}, \boldsymbol{\theta}' \in \mathbb{R}^d$ and $\boldsymbol{\omega}, \boldsymbol{\omega}' \in \mathbb{R}^d$, there exists a constant $L_f > 0$ such that $\|\nabla \mathcal{L}_i(\boldsymbol{\theta}, \boldsymbol{\omega}) - \nabla \mathcal{L}_i(\boldsymbol{\theta}', \boldsymbol{\omega}')\| \leq L_f (\|\boldsymbol{\theta} - \boldsymbol{\theta}'\| + \|\boldsymbol{\omega} - \boldsymbol{\omega}'\|)$.

---

**Algorithm 1** The Variance-Reduced Primal-Dual Stochastic Gradient Method (VRPD).

---

**Input:** An $M$-step trajectory of the state-action pairs $\{s_1, a_1, s_2, a_2, \cdots, s_M, a_M, s_{M+1}\}$ generated from a given policy; step sizes $\alpha, \beta \geq 0$; initialization points $\boldsymbol{\theta}^0 \in \mathbb{R}^d, \boldsymbol{\omega}^0 \in \mathcal{W}$.

**Output:** $(\boldsymbol{\theta}^{(\widetilde{K})}, \boldsymbol{\omega}^{(\widetilde{K})})$, where $\widetilde{K}$ is independently and uniformly picked from $\{1, \cdots, K\}$;

1: **for** $k = 0, 1, 2, \cdots, K - 1$ **do**
2:     If $\mathrm{mod}(k, q) = 0$, compute full gradients $G_{\boldsymbol{\theta}}^{(k)}, G_{\boldsymbol{\omega}}^{(k)}$ as in Eq. (6).
3: Otherwise, select $S$ samples independently and uniformly from $[M]$, and compute gradients as in Eq. (7).
4:     Perform the primal-dual updates to obtain the next iterate $\boldsymbol{\theta}^{(k+1)}, \boldsymbol{\omega}^{(k+1)}$ as in Eq. (8).
5: **end for**

---

*Assumption* 3 (Bounded Variance). There exists a constant $\sigma > 0$ such that for all $\boldsymbol{\theta} \in \mathbb{R}^d, \boldsymbol{\omega} \in \mathbb{R}^d$, $\frac{1}{M}\sum_{i=1}^{M}\|\nabla_{\boldsymbol{\theta}}\boldsymbol{\mathcal{L}}_i(\boldsymbol{\theta}, \boldsymbol{\omega}) - \nabla_{\boldsymbol{\theta}}\boldsymbol{\mathcal{L}}(\boldsymbol{\theta}, \boldsymbol{\omega})\|^2 \leq \sigma^2$ and $\frac{1}{M}\sum_{i=1}^{M}\|\nabla_{\boldsymbol{\omega}}\boldsymbol{\mathcal{L}}_i(\boldsymbol{\theta}, \boldsymbol{\omega}) - \nabla_{\boldsymbol{\omega}}\boldsymbol{\mathcal{L}}(\boldsymbol{\theta}, \boldsymbol{\omega})\|^2 \leq \sigma^2$.

In the above assumptions, Assumption 1 is satisfied if the number of samples $M$ is sufficiently large and coupling with the fact that the matrix $\boldsymbol{D}$ is positive definite. To see that, note that $\mu = \lambda_{\min}(\boldsymbol{D}) > 0$, where $\boldsymbol{D} = \mathbb{E}_s\left[\nabla_\theta V_\theta(s)\nabla_\theta V_\theta(s)^\top\right] \in \mathbb{R}^{d \times d}$ and $\boldsymbol{D}$ tends to be full-rank as $M$ increases. Thus, as soon as we find a $\mu > 0$ when $M$ is sufficiently large, this $\mu$ is independent of $M$ as $M$ continues to increase. Assumption 2 is standard in the optimization literature. Assumption 3 is also commonly adopted for proving convergence results of SGD- and VR-based algorithms, or algorithms that draw a mini-batch of samples instead of all samples. Assumption 3 is guaranteed to hold under the compact set condition and common for stochastic approximation algorithms for minimax optimization (Qiu et al., 2020; Lin et al., 2020a). Assumptions 1–3 are also general assumptions often used in temporal difference (TD) problems (see, e.g., (Qiu et al., 2020; Wai et al., 2019)). With these assumptions, we are now in a position to present our algorithms and their convergence performance results.

## 4.1 THE VARIANCE-REDUCED PRIMAL-DUAL METHOD

In this section, we first present the variance-reduced primal-dual (VRPD) algorithm for solving policy evaluation problems, followed by the theoretical convergence results. Due to space limitation, we provide a proof sketch in the main text and relegate the proof to the supplementary material.

**1) Algorithm Description:** The full description of VRPD is illustrated in Algorithm 1. In VRPD, for every $q$ iterations, the algorithm calculates the full gradients as follows:

$$G_{\boldsymbol{\theta}}^{(k)} = \frac{1}{|M|}\sum_{i \in M}\nabla_{\boldsymbol{\theta}}\boldsymbol{\mathcal{L}}_i(\boldsymbol{\theta}^{(k)}, \boldsymbol{\omega}^{(k)}); \quad G_{\boldsymbol{\omega}}^{(k)} = \frac{1}{|M|}\sum_{i \in M}\nabla_{\boldsymbol{\omega}}\boldsymbol{\mathcal{L}}_i(\boldsymbol{\theta}^{(k)}, \boldsymbol{\omega}^{(k)}). \tag{6}$$

In all other iterations, VRPD selects a batch of samples $S$ and computes variance-reduced gradient estimators as:

$$G_{\boldsymbol{\theta}}^{(k)} = \frac{1}{|S|}\sum_{i \in S}\left(\nabla_{\boldsymbol{\theta}}\boldsymbol{\mathcal{L}}_i(\boldsymbol{\theta}^{(k)}, \boldsymbol{\omega}^{(k)}) - \nabla_{\boldsymbol{\theta}}\boldsymbol{\mathcal{L}}_i(\boldsymbol{\theta}^{(k-1)}, \boldsymbol{\omega}^{(k-1)}) + G_{\boldsymbol{\theta}}^{(k-1)}\right); \tag{7a}$$

$$G_{\boldsymbol{\omega}}^{(k)} = \frac{1}{|S|}\sum_{i \in S}\left(\nabla_{\boldsymbol{\omega}}\boldsymbol{\mathcal{L}}_i(\boldsymbol{\theta}^{(k)}, \boldsymbol{\omega}^{(k)}) - \nabla_{\boldsymbol{\omega}}\boldsymbol{\mathcal{L}}_i(\boldsymbol{\theta}^{(k-1)}, \boldsymbol{\omega}^{(k-1)}) + G_{\boldsymbol{\omega}}^{(k-1)}\right). \tag{7b}$$

The estimators in (7) are constructed iteratively based on the previous update information $\nabla_{\boldsymbol{\theta}}\boldsymbol{\mathcal{L}}_i(\boldsymbol{\theta}^{(k-1)}, \boldsymbol{\omega}^{(k-1)})$ (resp. $(\nabla_{\boldsymbol{\omega}}\boldsymbol{\mathcal{L}}_i(\boldsymbol{\theta}^{(k-1)}, \boldsymbol{\omega}^{(k-1)})$ ) and $G_{\boldsymbol{\theta}}^{(k-1)}$ (resp. $G_{\boldsymbol{\omega}}^{(k-1)}$). VRPD updates the primal and dual variables as follows:

$$\boldsymbol{\theta}^{(k+1)} = \boldsymbol{\theta}^{(k)} - \beta G_{\boldsymbol{\theta}}^{(k)}; \tag{8a}$$

$$\boldsymbol{\omega}^{(k+1)} = \mathcal{P}_{\mathcal{W}}(\boldsymbol{\omega}^{(k)} + \alpha G_{\boldsymbol{\omega}}^{(k)}) = \underset{\widetilde{\boldsymbol{\omega}} \in \Omega}{\mathrm{argmin}}\|\widetilde{\boldsymbol{\omega}} - (\boldsymbol{\omega}^{(k)} + \alpha G_{\boldsymbol{\omega}}^{(k)})\|^2, \tag{8b}$$

where the parameters $\alpha$ and $\beta$ are constant learning rates for primal and dual updates, respectively.

**2) Convergence Performance:** In this paper, we propose a *new metric* for convergence analysis:

$$\mathfrak{M}^{(k)} := \|\nabla J(\boldsymbol{\theta}^{(k)})\|^2 + 2\|\boldsymbol{\omega}^{(k)} - \boldsymbol{\omega}^*(\boldsymbol{\theta}^{(k)})\|^2. \tag{9}$$

The first term in (9) measures the convergence of the primal variable $\boldsymbol{\theta}$. As common in non-convex optimization analysis, $\|\nabla J(\boldsymbol{\theta})\|^2 = 0$ indicates that $\boldsymbol{\theta}$ is a first-order stationary point

(FOSP) of Problem (4). The second term in (9) measures the convergence of $\boldsymbol{\omega}^{(k)}$ to the unique maximizer $\boldsymbol{\omega}^{(k)}*$ for $\mathcal{L}(\boldsymbol{\theta}_k, \cdot)$. Note that if Problem (4) is unconstrained in dual (i.e., $\boldsymbol{\omega} \in \mathbb{R}^d$), it follows from Assumption 2 and $\|\nabla_{\boldsymbol{\omega}} \mathcal{L}(\boldsymbol{\theta}^{(k)}, \boldsymbol{\omega}^*(\boldsymbol{\theta}^{(k)}))\|^2 = 0$ that $\mathfrak{M}^{(k)} \geq \|\nabla J(\boldsymbol{\theta}^{(k)})\|^2 + \frac{2}{L_f^2}\|\nabla_{\boldsymbol{\omega}} \mathcal{L}(\boldsymbol{\theta}^{(k)}, \boldsymbol{\omega}^{(k)})\|^2$. We now introduce the notion of the *approximate first-order stationary points*. We say that point $\{\boldsymbol{\theta}, \boldsymbol{\omega}\}$ is an $\epsilon$-stationary point of function $\mathcal{L}(\boldsymbol{\theta}, \boldsymbol{\omega})$ if $\mathfrak{M} \leq \epsilon$ is satisfied.

**Remark.** Several important remarks on the connections between our metric $\mathfrak{M}^{(k)}$ and the conventional convergence metrics in the literature are in order. A conventional convergence metric in the literature for NCSC minimax optimization is $\|\nabla J(\boldsymbol{\theta}^{(k)})\|^2$ (Lin et al., 2020a; Luo et al., 2020; Zhang et al., 2021), which is the first term of $\mathfrak{M}^{(k)}$ and measures the convergence of the primal variable $\boldsymbol{\theta}$ under a given dual variable $\boldsymbol{\omega}$. This is because $\|\nabla J(\boldsymbol{\theta})\|^2 = 0$ implies that $\boldsymbol{\theta}$ is a FOSP. The *novelty* in our convergence metric is the second term in $\mathfrak{M}^{(k)}$, which measures the convergence of $\boldsymbol{\omega}_k$ to the unique maximizer $\boldsymbol{\omega}_k^*$ for $\mathcal{L}(\boldsymbol{\theta}_k, \cdot)$.

Another conventional convergence metric in the literature of minimizing the empirical MSPBE problem is $\|\nabla_{\boldsymbol{\theta}} \mathcal{L}(\boldsymbol{\theta}, \boldsymbol{\omega})\|^2 + \|\nabla_{\boldsymbol{\omega}} \mathcal{L}(\boldsymbol{\theta}, \boldsymbol{\omega})\|^2$ (Tsitsiklis & Van Roy, 1997). Since the nonconvex-strong-concave minimax optimization problem is unconstrained in dual (i.e., $\boldsymbol{\omega} \in \mathbb{R}^d$), it follows from Lipschitz-smoothness in Assumption 2 and $\|\nabla_{\boldsymbol{\omega}} \mathcal{L}(\boldsymbol{\theta}^{(k)}, \boldsymbol{\omega}^*(\boldsymbol{\theta}^{(k)}))\|^2 = 0$ that $\|\boldsymbol{\omega}^{(k)} - \boldsymbol{\omega}^*(\boldsymbol{\theta}^{(k)})\|^2 \geq \frac{1}{L_f^2}\|\nabla_{\boldsymbol{\omega}} \mathcal{L}(\boldsymbol{\theta}^{(k)}, \boldsymbol{\omega}^{(k)})\|^2$. Therefore, the second term in our $\mathfrak{M}^{(k)}$ $(2\|\boldsymbol{\omega}^{(k)} - \boldsymbol{\omega}^*(\boldsymbol{\theta}^{(k)})\|^2)$ is an upper bound of the second term in this conventional metric $(\|\nabla_{\boldsymbol{\omega}} \mathcal{L}(\boldsymbol{\theta}, \boldsymbol{\omega})\|^2)$. Thus, $2\|\boldsymbol{\omega}^{(k)} - \boldsymbol{\omega}^*(\boldsymbol{\theta}^{(k)})\|^2$ is a *stronger* metric than $\|\nabla_{\boldsymbol{\omega}} \mathcal{L}(\boldsymbol{\theta}, \boldsymbol{\omega})\|^2$ in the sense that an $O(1/K)$ convergence rate under $\mathfrak{M}^{(k)}$ implies an $O(1/K)$ convergence rate of the conventional metric, but the converse is *not* true. Moreover, the benefit of using $2\|\boldsymbol{\omega}^{(k)} - \boldsymbol{\omega}^*(\boldsymbol{\theta}^{(k)})\|^2$ in our $\mathfrak{M}^{(k)}$ is that its special structure allows us to prove the $O(1/K)$ convergence, while the second term in the conventional metric fails. $\square$

With our proposed convergence metric in (9), we have the following convergence result:

**Theorem 1.** Under Assumptions 1–3, choose step-sizes: $\alpha \leq \min\{\frac{1}{4L_f}, \frac{2\mu}{34L_f^2 + 2\mu^2}\}$ and $\beta \leq \min\left\{\frac{1}{4L_f}, \frac{1}{2(L_f + L_f^2/\mu)}, \frac{\mu}{8\sqrt{17}L_f^2}, \frac{\mu^2\alpha}{8\sqrt{34}L_f^2}\right\}$. Let $q = \sqrt{M}$ and $S = \sqrt{M}$, it holds that:

$$\frac{1}{K}\sum_{k=0}^{K-1}\mathbb{E}[\mathfrak{M}^{(k)}] \leq \frac{1}{K\min\{1, L_f^2\}}\left[\frac{16L_f^2}{\alpha\mu}C_2 + \frac{2}{\beta}C_1\right] = \mathcal{O}\left(\frac{1}{K}\right),$$

where $C_1 \triangleq \mathbb{E}[J(\boldsymbol{\theta}^{(0)})] - \mathbb{E}[J(\boldsymbol{\theta}^{(*)})]$ and $C_2 \triangleq (\mathbb{E}\|\boldsymbol{\omega}^*(\boldsymbol{\theta}^{(0)}) - \boldsymbol{\omega}^{(0)}\|^2)$.

**Corollary 2.** The overall stochastic sample complexity is $\mathcal{O}(\sqrt{M}\kappa^3\epsilon^{-1} + M)$. Note that $\kappa = L_f/\mu$ denotes the condition number.

**Remark.** Theorem 1 states that VRPD achieves an $\mathcal{O}(1/K)$ convergence rate to an $\epsilon$-FOSP. The most challenging part in proving Theorem 1 stems from the fact that one needs to simultaneously evaluate the progresses of the gradient descent in the primal domain and the gradient ascent in the dual domain of the minimax problem.

Toward this end, the nPD-VR method in (Wai et al., 2019) employs $\|\nabla_{\boldsymbol{\omega}} \mathcal{L}(\boldsymbol{\theta}^{(k)}, \boldsymbol{\omega}^{(k)})\|^2$ in their metric to evaluate convergence. However, this approach yields a term $F^{(K)} \triangleq \mathbb{E}[\mathcal{L}(\boldsymbol{\theta}^{(0)}, \boldsymbol{\omega}^{(0)}) - \mathcal{L}(\boldsymbol{\theta}^{(K)}, \boldsymbol{\omega}^{(K)})]$ in their convergence upper bound in the form of $\mathcal{O}(F^{(K)}/K)$ (cf. Theorem 1, Eq. (26) in (Wai et al., 2019)). Since $F^{(K)}$ depends on $K$, it is unclear whether or not the nPD-VR method in (Wai et al., 2019) can achieve an $\mathcal{O}(1/K)$ convergence rate. This unsatisfactory result motivates us to propose a new metric $\mathfrak{M}^{(k)}$ in Eq. (9) to evaluate the convergence of our VRPD algorithm. The first part of our convergence metric $\|\nabla J(\boldsymbol{\theta}^{(k)})\|^2$ measures the *stationarity gap* of the primal variable, while the second part $2\|\boldsymbol{\omega}^{(k)} - \boldsymbol{\omega}^*(\boldsymbol{\theta}^{(k)})\|^2$ measures the dual *optimality gap*. Consequently, we bound per-iteration change in $J(\boldsymbol{\theta})$ instead of the function $\mathcal{L}(\boldsymbol{\theta}^{(k)}, \boldsymbol{\omega}^{(k)})$. This helps us avoid the technical limitations of (Wai et al., 2019) and successfully establish the $\mathcal{O}(1/K)$ convergence rate, hence resolving an open problem in this area. $\square$

**Remark.** VRPD adopts a large $\mathcal{O}(1)$ (i.e., constant) step-size compared to the $\mathcal{O}(1/M)$ step-size of nPD-VR (Wai et al., 2019), where $M$ is the dataset size. This also induces a faster convergence. Also, VRPD's estimator uses fresher information from the previous iteration, while VR-STSG (Qiu

---

**Algorithm 2** Adaptive-batch VRPD method (VRPD$^+$).

---

**Input:** A trajectory of the state-action pairs $\{s_1, a_1, s_2, a_2, \cdots, s_M, a_M, s_{M+1}\}$ generated from a given policy; step sizes $\alpha, \beta \geq 0$; initialization points $\boldsymbol{\theta}^0 \in \Theta, \boldsymbol{\omega}^0 \in \mathbb{R}^d$.

**Output:** $(\boldsymbol{\theta}^{(\widetilde{K})}, \boldsymbol{\omega}^{(\widetilde{K})})$, where $\widetilde{K}$ is independently and uniformly picked from $\{1, \cdots, K\}$;

1: **for** $k = 0, 1, 2, \cdots, K - 1$ **do**
2:     If $\mathrm{mod}(k, q) = 0$, select $\mathcal{N}_s$ indices independently and uniformly from $[M]$ as in Eq. (10) and calculate stochastic gradients as in Eq. (11);
3:     Otherwise, select $S$ independently and uniformly from $[M]$; Compute gradients as in Eq. (7);
4:     Perform the primal-dual updates as in Eq. (8).
5: **end for**

---

et al., 2020) and nPD-VR (Wai et al., 2019) only use the information from the beginning of $q$-sized windows. Collectively, VRPD makes a considerably larger progress than state-of-the-art algorithms (Qiu et al., 2020; Wai et al., 2019). □

### 4.2 THE ADAPTIVE-BATCH VRPD METHOD (VRPD$^+$)

Note that VRPD still requires full gradients every $q$ iterations, which may entail a high sample complexity. Upon closer observations, we note that accurate gradient estimation plays an important role *only* in the later stage of the convergence process. This motivates us to further lower the sample complexity of VRPD by using *adaptive batch sizes*. Toward this end, we propose an adaptive-batch VRPD method (VRPD$^+$) to lower the sample complexity of the VRPD algorithm in Algorithm 1.

**1) Algorithm Description:** The full description of VRPD$^+$ is illustrated in Algorithm 2. In VRPD$^+$, our key idea is to use the gradients calculated in the previous loop to adjust the batch size $\mathcal{N}_s$ of the next loop. Specifically, VRPD$^+$ chooses $\mathcal{N}_s$ in the $k$-th iteration as:

$$\mathcal{N}_s = \min\{c_\gamma \sigma^2 (\gamma^{(k)})^{-1}, c_\epsilon \sigma^2 \epsilon^{-1}, M\}, \tag{10}$$

where $c_\gamma, c_\epsilon > c$ for certain constant $c$, $M$ denotes the size of the dataset and $\sigma^2$ is the variance bound, and $\gamma^{(k+1)} = \sum_{i=(n_k-1)q}^{k} \frac{\|G_{\boldsymbol{\theta}}^{(i)}\|^2}{q}$ is the stochastic gradients calculated in the previous iterations. In VRPD$^+$, for every $q$ iterations, we select $\mathcal{N}_s$ samples independently and uniformly from $[M]$ and compute gradient estimators as follows:

$$G_{\boldsymbol{\theta}}^{(k)} = \frac{1}{|\mathcal{N}_s|} \sum_{i \in \mathcal{N}_s} \nabla_{\boldsymbol{\theta}} \boldsymbol{\mathcal{L}}_i(\boldsymbol{\theta}^{(k)}, \boldsymbol{\omega}^{(k)}); \quad G_{\boldsymbol{\omega}}^{(k)} = \frac{1}{|\mathcal{N}_s|} \sum_{i \in \mathcal{N}_s} \nabla_{\boldsymbol{\theta}} \boldsymbol{\mathcal{L}}_i(\boldsymbol{\theta}^{(k)}, \boldsymbol{\omega}^{(k)}). \tag{11}$$

For other iterations, VRPD$^+$ is exactly the same as VRPD. Next, we will theoretically show that such an adaptive batch-size scheme still retains the same convergence rate, while achieving an improved sample complexity.

**2) Convergence Performance:** For VRPD$^+$, we have the following convergence performance result:

**Theorem 3.** Under Assumptions 1–3, choose step-sizes: $\alpha \leq \min\{\frac{1}{4L_f}, \frac{2\mu}{34L_f^2 + 2\mu^2}\}$ and $\beta \leq \min\left\{\frac{1}{4L_f}, \frac{1}{2(L_f + L_f^2/\mu)}, \frac{\mu}{8\sqrt{17}L_f^2}, \frac{\mu^2\alpha}{8\sqrt{34}L_f^2}\right\}$. Let $q = \sqrt{M}, S = \sqrt{M}$ and $c_\gamma \geq (288L_f^2/\mu^2 + 8)$ in VRPD$^+$, where $c_\gamma \geq c$ for some constant $c > 4K + \frac{68K}{\beta\mu^2}$. With constants $C_1 \triangleq \mathbb{E}[J(\boldsymbol{\theta}^{(0)})] - \mathbb{E}[J(\boldsymbol{\theta}^{(*)})]$ and $C_2 \triangleq \left(\mathbb{E}[\|\boldsymbol{\omega}^*(\boldsymbol{\theta}^{(0)}) - \boldsymbol{\omega}^{(0)}\|^2]\right)$, it holds that:

$$\frac{1}{K} \sum_{k=0}^{K-1} \mathbb{E}[\mathfrak{M}^{(k)}] \leq \frac{1}{K\min\{1, L_f^2\}}\left[K \cdot \frac{\epsilon}{2} + \frac{16L_f^2}{\alpha\mu}C_2 + \frac{2}{\beta}C_1\right] = \mathcal{O}\left(\frac{1}{K}\right) + \frac{\epsilon}{2}.$$

**Corollary 4.** The overall stochastic sample complexity is $\mathcal{O}(\sqrt{M}\kappa^3\epsilon^{-1} + M)$. $\kappa = L_f/\mu$ denotes the condition number.

**Remark.** From Theorem 3, it can be seen that VRPD$^+$ achieves the same convergence rate as that of VRPD. Since we choose the subsample set $\mathcal{N}_s$ instead of full gradient calculation in VRPD$^+$, it achieves a much lower sample complexity compared to VRPD. Additionally, the convergence performance of VRPD$^+$ is affected by the constant $\frac{K\epsilon}{2}$, which is due to the use of the adaptive batch size in each outer-loop of VRPD$^+$. Also, it can be observed that the algorithm convergence rate is affected by the carefully chosen step-sizes $\alpha$ and $\beta$, because either a too small or too large step-size may have negative impact on the convergence of the algorithm. □

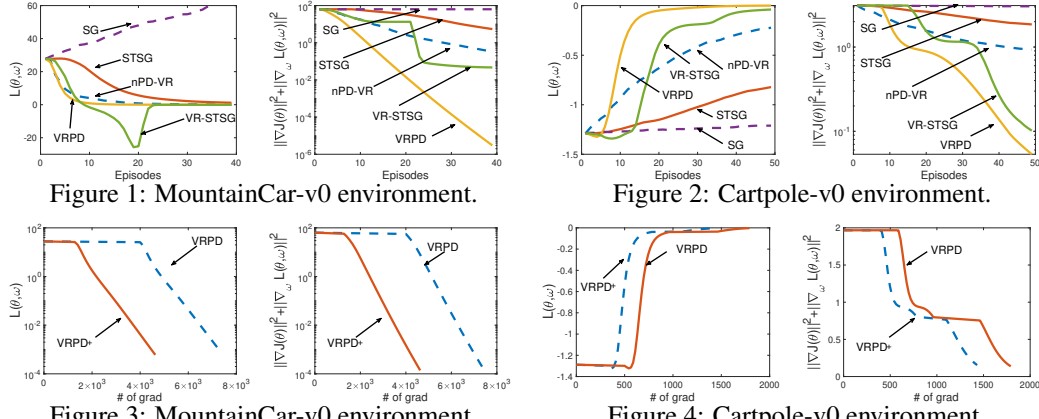

Figure 1: MountainCar-v0 environment.   Figure 2: Cartpole-v0 environment.

Figure 3: MountainCar-v0 environment.   Figure 4: Cartpole-v0 environment.

**Remark.** The proof of Theorem 3 follows from a similar approach to the proof of Theorem 1. The key difference and most challenging part of proving Theorem 3 stem from the relaxation on $\|\nabla_{\boldsymbol{\theta}}\mathcal{L}(\boldsymbol{\theta}^{(k)}, \boldsymbol{\omega}^{(k)}) - G_{\boldsymbol{\theta}}^{(k)}\|^2$ and $\|\nabla_{\boldsymbol{\omega}}\mathcal{L}(\boldsymbol{\theta}^{(k)}, \boldsymbol{\omega}^{(k)}) - G_{\boldsymbol{\omega}}^{(k)}\|^2$. Thanks to the bounded variance in Assumption 3 and the selected $\mathcal{N}_s$ in Eq. (10), we are able to derive outer-loop bounds for primal and dual gaps, respectively. We refer readers to the Appendix for the details of the complete proof. $\square$

## 5   EXPERIMENTAL RESULTS

In this section, we conduct numerical experiments to verify our theoretical results. We compare our work with the basic stochastic gradient (SG) method (Lin et al., 2020b) and three state-of-the-art algorithms for PE: nPD-VR (Wai et al., 2019), STSG (Qiu et al., 2020) and VR-STSG (Qiu et al., 2020). Due to space limitation, we provide our detailed experiment setting in the Appendix.

**Numerical Results:** We set the constant learning rates $\alpha = 10^{-3}, \beta = 10^{-1}$, mini-batch size $q = \lceil\sqrt{M}\rceil$, constant $c = 32$ and solution accuracy $\epsilon = 10^{-3}$. First, we compare the loss value and gradient norm performance based on MountainCar-v0 and Cartpole-v0 with nPD-VR, SG, STSG, and VR-STSG in Figs. 1 and 2. We set the constraint $\mathcal{W} = [0, 10]^n$ and initialize all algorithms at the same point, which is generated randomly from the normal distribution. We can see that VR-STSG and nPD-VR slowly converge after 40 epochs, while STSG and SG fail to converge. VRPD converges faster than all the other algorithms with the same step-size values. As for Cartpole-v0, we clearly see a trend of approaching zero-loss with VRPD. These results are consistent with our theoretical result that one can use a relatively large step-size with VRPD, which leads to faster convergence. Also, we compare the sample complexity of VRPD and VRPD$^+$ in MountainCar-v0 and Cartpole-v0, and the results are shown in in Figs. 3 and 4, respectively. We can see

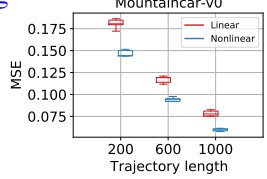

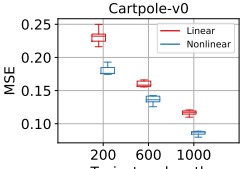

Figure 5: MSE comparison with 10 trials.

that VRPD$^+$ converges to the same level with much fewer samples than VRPD does. Next, we compared the mean squared error(MSE) between the ground truth value function and the estimated value function over 10 independent runs with linear approximation and nonlinear approximation. In Fig. 5, with the same amount of parameter size, nonlinear approximation always achieves smaller MSE than linear approximation (Du et al., 2017). Further experiments on the performance of $J(\boldsymbol{\theta})$ are shown in the supplementary material.

## 6   CONCLUSION

In this paper, we proposed and analyzed two algorithms called VRPD and VRPD$^+$ for policy evaluation with nonlinear approximation. The VRPD algorithm is based on a simple single-timescale framework by utilizing variance reduction techniques. The VRPD algorithm allows the use of constant step-sizes and achieves an $\mathcal{O}(1/K)$ convergence rate. The VRPD$^+$ algorithm improves VRPD by further applying an adaptive batch size based on historical stochastic gradient information. Our experimental results also confirmed our theoretical findings in convergence and sample complexity.

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
