# OpenReview forum: "On $\mathcal{O}(1/K)$ Convergence and Low Sample Complexity for Single-Timescale Policy Evaluation with Nonlinear Function Approximation"
_ICLR.cc/2023/Conference — Submitted to ICLR 2023_

### Official Review · Reviewer_iPgL · 2022-10-22

**Confidence:** 2
**Correctness:** 4
**Technical Novelty And Significance:** 3
**Empirical Novelty And Significance:** Not applicable
**Recommendation:** 6

**Clarity, Quality, Novelty And Reproducibility:**

The paper is of high quality and clarity.

Questions:
1. Regarding the weakness mentioned above, can you establish guarantees for the two terms in Eqn (9) separately?
2. It seems to sample a trajectory with a length of $M$ cannot be used as a finite-sample approximation of the stationary distribution. In discounted setting, the classical way to do so is to run the trajectory with a stopping probability of $1-\gamma$, and only use the last step as an unbiased sample from the stationary distribution. This doesn't really affect your optimization results but should be made clear under the context of RL or policy evaluation.

**Strength And Weaknesses:**

Strength:
1. Paper is well-written and easy to follow.
2. Compared with previous work on the non-convex-strongly-concave minimax optimization problem, the convergence rate of this work achieves $O(1/K)$, which is the same as before and has fewer constraints.

Weakness:
1. The convergence metric is unconventional. The first term in Eqn (9) is the full gradient w.r.t. $\theta$, which (I believe) includes both the partial derivative of the first input and the chained gradient w.r.t. $\theta$ via the second input. This first term already suffices to guarantee a first-order stationary point, rendering the second term redundant. On the other hand, bounding the partial gradient of both inputs is also acceptable. I wonder why the analysis of this paper fails to provide a guarantee for either of these metrics.
2. The technical novelty of the minimax optimization technique used here is not clearly stated and compared with previous works. I believe it helps to also make a list to compare with previous NCSC minimax optimization algorithms.

**Summary Of The Paper:**

This paper studies policy evaluation with nonlinear function approximation. It proposes and analyzed two algorithms called VRPD and VRPD+ to optimize the primal-dual form of the MSBPE. The VRPD algorithm utilizes variance reduction techniques to achieve $O(1/K)$ convergence rate.

**Summary Of The Review:**

Currently, the paper shows strong theoretical results compared with previous literature. Some concerns still need to be addressed. Thus I recommend weak acceptance.

---

> ### Author Response · Authors · 2022-11-19
> **Response to Reviwer iPgL's Comments [Part 1]**
>
>
> > **Your Comment 1:** The convergence metric is unconventional. The first term in Eqn (9) is the full gradient w.r.t. , which (I believe) includes both the partial derivative of the first input and the chained gradient w.r.t.  via the second input. This first term already suffices to guarantee a first-order stationary point, rendering the second term redundant. On the other hand, bounding the partial gradient of both inputs is also acceptable. I wonder why the analysis of this paper fails to provide a guarantee for either of these metrics.
>
> **Our Response:** Thanks for your comments, we agree with the reviewer that the first term $||\nabla J(\theta^{(k)}) ||^2 \rightarrow 0$ in our convergence metric already suffices to guarantee a first-order stationary point. However, using the first term alone implicitely assume that we have the optimal dual solution $\omega^*(\theta^k)$ for any given $\theta^k$-value, which typically needs to be achieved asymptotically (i.e., by a two-timescale algorithm that takes an infinite number of steps between each successive iterations in $k$) except for some problems with special structures. Such an assumption of having the $\omega^*(\theta^k)$-solution is hard to satisfy for most algorithms in practice.
>
> Thus, to characterize the *finite-time convergence* of both the primal and dual variables in our primal-dual algorithm for any given iteration $k$, we added the second term in our convergence metric. Clearly, our metric  $|| \nabla J (\theta^{(k)}) ||^2 +||\omega^{(k)}-\omega^{*}(\theta^{(k)}) ||^2$
>
>  is stronger than using $||\nabla J(\theta^{(k)}) ||^2$ alone since our metric further measures the convergence of the dual variables $\omega^{(k)}$ to the unique maximizer $\omega^{(k)}*$ for $\mathcal L(\theta_k,\cdot)$.
>
>
>
> > **Your Comment 2:** The technical novelty of the minimax optimization technique used here is not clearly stated and compared with previous works. I believe it helps to also make a list to compare with previous NCSC minimax optimization algorithms.
>
> **Our Response:** Thanks for the suggestion. Due to the space limitation, we weren't able to add a list to compare with previous NCSC min-max algorithms at the time of submission. In this revision, we have added the NCSC algorithm comparisons in our Appendix. The major technical novelty of our algorithms compared to other NCSC algorithms are already discussed in Section 2 and Section 4. Here, we provide a quick summary of our technical novelty:
>
> * First, our proposed algorithm VRPD adopts a large $\mathcal{O}(1)$ (i.e., constant) step-size compared to the $\mathcal{O}(1/M)$ step-size of nPD-VR [Wai et al., 2019], where $M$ is the dataset size, which also induces a faster convergence.
>
> * Second, VRPD's estimator uses fresher information from the previous iteration, while VR-STSG [Qiu et al., 2020] and nPD-VR [Wai et al., 2019] only use the information from the beginning of $q$-sized windows.
>
> * Third, our algorithms are primal-dual algorithms that update the two variables simultaneously rather than alternatively. Thus, our proposed algorithms are much simpler and significantly different from SREDA, requiring solving the maximization subproblem for the dual variable at each iteration.
>
> > **Your Comment 3:** Regarding the weakness mentioned above, can you establish guarantees for the two terms in Eqn (9) separately?
>
> **Our Response:** Yes. Since both $||\nabla J(\theta^{(k)})||^2$ and $||\omega^{(k)}-\omega^{*}(\theta^{(k)}) ||^2$
>
> are two non-negative terms, we can immediately conclude that $|| \nabla J(\theta^{(k)}) ||^2$ and $|| \omega^{(k)}-\omega^{*}(\theta^{(k)}) ||^2$ both have an $\mathcal{O}(1/K)$ convergence rate once we establish the $\mathcal{O}(1/K)$ convergence rate of our convergence metric.

---

> > ### Author Response · Authors · 2022-11-19
> > **Response to Reviwer iPgL's Comments [Part 2]**
> >
> >
> > > **Your Comment 4:**  It seems to sample a trajectory with a length of $M$ cannot be used as a finite-sample approximation of the stationary distribution. In discounted setting, the classical way to do so is to run the trajectory with a stopping probability of $1-\gamma$, and only use the last step as an unbiased sample from the stationary distribution. This doesn't really affect your optimization results but should be made clear under the context of RL or policy evaluation.
> >
> > **Our Response:** Thanks for insightful comments. We fullly agree with the reviewer's point and will add the clarification as the reviewer suggested!
> >
> > Just for the fun of further academic discussions, in here, we also want to point out that the use of an $M$-trajectory is actually not uncommon in the literature. For example, the finite-sum min-max objective function based on an $M$-trajectory has been used in Eq.(10) in [Wai et al. 2019], Eq.(3) in [Qiu et al. 2020], and [Liu, et al. JAIR’18]). In what follows, we would also like to briefly describe two cases where this approach may still serve as a good approximation, since the resultant bias of the finite-sum approach based on an $M$-trajectory can be controlled or bounded by increasing the numbers of time steps and batch size.
> >
> > * First, the underlying Markov decision process (MDP) of the RL system is "fast mixing" if it satisfies the following assumption of the ergodicity property (a widely used assumption in the literature of convergence analysis of finite-sample analysis of RL algorithms, see [R4] and references therein):
> >
> > $\sup _{s\in \mathcal{S}} || \mathbb{P}(s _t \in \cdot|s _0 = s) - \chi _\pi || _{TV} \leq \kappa \rho^t$, $\forall t\geq 0$,
> >
> > where $\kappa>0$ and $0<\rho<1$ are constants, $\|\cdot\|_ {TV}$ represents total variation, $\pi$ is the policy being considered, and $\chi_{\pi}$ is the stationary distribution of the corresponding MDP with transition kernel $\mathbb{P}(\cdot|s,a)$ under policy $\pi$. Note that this ergodicity property always holds for time-homogeneous Markov chain with finite state space or any uniformly ergodic Markov chain with general state space. Thanks to this "fast mixing" nature, the trajectories quickly approach the stationary distribution as the number of time slots $T$ and the batch size of the trajectories $N$ increase. Thus, one can obtain close-to-stationary samples by discarding a portion of the initial transient samples.
> >
> > * Second, some existing works (e.g., [R5]) have also analyzed and quantified the bias resulted from using finite-sum with finite samples from an $M$-trajectory. For example, in [R5], the authors showed that if i) the loss (or negative of reward) of any state-action pair is bounded (see Assumption 1 in [R5]) and ii) the score function its partial derivatives are bounded (see Assumption 2 in [R5]) (both of which are not restrictive in practice), then the bias resulted from non-stationarity can be bounded (the policy gradient concentration result in Lemma 2 in [R5]).
> >
> > To summarize, under the above circumstances, the use of the empirical finite-sum objective based on an $M$-trajectory may be considered acceptable. But again, we will make the clarifications as the reviewer suggested.
> >
> >
> >
> > [R4] Xu, Tengyu, Zhe Wang, and Yingbin Liang. "Non-asymptotic convergence analysis of two time-scale (natural) actor-critic algorithms." arXiv preprint arXiv:2005.03557 (2020)
> >
> > [R5] Chen, Tianyi, et al. "Communication-efficient policy gradient methods for distributed reinforcement learning." IEEE Transactions on Control of Network Systems 9.2 (2021): 917-929.

---

### Official Review · Reviewer_ZMya · 2022-10-24

**Confidence:** 4
**Correctness:** 2
**Technical Novelty And Significance:** 1
**Empirical Novelty And Significance:** 2
**Recommendation:** 3

**Clarity, Quality, Novelty And Reproducibility:**

The paper is poorly organized and hard to follow. There are too many unjustified and misleading claims.

**Strength And Weaknesses:**

Strength:

- The paper proposes a variance reduction method for solving a problem that seems to have a connection with the policy evaluation problem. They obtain the $O(\frac{1}{K})$ iteration complexity.

Weakness:
- The considered finite-sum objective differs from the original MSPBE objective. In fact, the objective cannot even be viewed as an unbiased substitute, as the sample from a trajectory doesn’t follow the stationary distribution. Such an objective is not justified.
- The algorithm’s update is poorly explained. From my view, it’s just SPIDER’s update [1], which is a popular variance reduction approach. However, there is no discussion with existing variance reduction methods.
- The projection step in (8) is not explained.
- There are too many typos and unprecise claims, making the paper hard to read. Here are examples:
1. The last $g_\theta(s)$ in (1) should not have a transpose.
2.  The strongly-concavity definition in assumption 4.2 is wrong; the function should also be additionally concave.
3.  The L_f-smoothness definition in assumption 4.3 is unprecise; theta and omega should be in the same norm.

[1] Spider: Near-Optimal Non-Convex Optimization via Stochastic Path Integrated Differential Estimator.

**Summary Of The Paper:**

The paper considers the policy evaluation problem of reinforcement learning. Based on a primal-dual formulation of the objective, the authors apply a variance reduction technique to solve the problem under the nonlinear function approximation. They claim that the algorithm attains $O(\frac{1}{K})$ iteration complexity.

**Summary Of The Review:**

The paper’s motivation is not clearly centered. The optimization objective is not justified and the algorithm is not explained. There are many typos and technical flaws which make the paper hard to understand. Thus, I believe the work is not finished and is far from its fine shape for publication in ICLR.

---

> ### Author Response · Authors · 2022-11-19
> **Response to Reviewer ZMya's Comments [Part 1]**
>
> > **Your Comment 1:** The considered finite-sum objective differs from the original MSPBE objective. In fact, the objective cannot even be viewed as an unbiased substitute, as the sample from a trajectory doesn't follow the stationary distribution. Such an objective is not justified.
>
>
> **Our Response:**  Thanks for your comments. We agree with the reviewer that the finite-sum objective is a biased substitute of the orginal MSPBE. However, as the pointed out by the reviewer, since the stationary distribution is unknown in RL, it is difficult to directly evaluate the original MSPBE objective. Instead, researchers have considered using an empirical finite-sum objective as a proxy (see our discussion on Page 5). In fact, the finite-sum min-max objective function for substituting the original MSPBE objective is a standard approach that is widely adopted in the literature for RL policy evaluation (see, e.g., Eq.(10) in [Wai et al. 2019], Eq.(3) in [Qiu et al. 2020], and and [Liu, et al. JAIR’18]). For fair performance comparison with these existing works, we also adopted the same finite-sum min-max objective function. Also, how to construct a better substituion for the original MSPBE objective function remains an open problem, which is important in its own right and deserves an independent paper. We will consider this important topic in our future study.
>
>
> Although multiple previous works have already justified the use of the empirical finite-sum objective (see, e.g., [Wai et al. 2019], [Qiu et al. 2020], and [Liu, et al. JAIR’18]), we would also like to briefly explain when and where the empirical finite-sum objective function could serve as a good approximation. It is true that, due to the Markovian nature of the state-action samples in RL, the finite-sum objective is not an unbiased estimate of the original MSPBE objective. However, in the following two cases, the resultant bias of the finite-sum approach can be controlled or bounded by increasing the numbers of time steps and batch size.
>
> * First, the underlying Markov decision process (MDP) is "fast mixing" if it satisfies the following assumption on the ergodicity property (a widely used assumption in the literature of convergence analysis of finite-sample analysis of RL algorithms, see [R4] and references therein): $\sup _{s \in \mathcal{S}} || \mathbb{P}(s _t \in \cdot|s_0 = s) - \chi _\pi || _{TV} \leq \kappa \rho^t$, $\forall t\geq 0$, where $\kappa>0$ and $0<\rho<1$ are constants, $||\cdot|| _{TV}$ represents total variation, $\pi$ is the policy being considered, and $\chi _{\pi}$ is the stationary distribution of the corresponding MDP with transition kernel $\mathbb{P}(\cdot|s,a)$ under policy $\pi$. Note that this ergodicity property always holds for time-homogeneous Markov chain with finite state space or any uniformly ergodic Markov chain with general state space. Thanks to this "fast mixing" nature, the trajectories quickly approach the stationary distribution as the number of time slots $T$ and the batch size of the trajectories $N$ increase. Thus, one can obtain close-to-stationary samples by discarding a portion of the initial transient samples.
>
> * Second, some existing works (e.g., [R5]) have also analyzed and quantified the bias resulted from using finite-sum with finite samples. For example, in [R5], the authors showed that if i) the loss (or negative of reward) of any state-action pair is bounded (see Assumption 1 in [R5]) and ii) the score function its partial derivatives are bounded (see Assumption 2 in [R5]) (both of which are not restrictive in practice), then the bias resulted from non-stationarity can be bounded (the policy gradient concentration result in Lemma 2 in [R5]).
>
> To summarize, under the above circumstances, the use of the empirical finite-sum objective can be considered acceptable. We will add the above discussion in the revision of this paper. We thank the reviewer for this insightful question, which definitely improves the clarity of this paper.
>
> [R4] Xu, Tengyu, Zhe Wang, and Yingbin Liang. "Non-asymptotic convergence analysis of two time-scale (natural) actor-critic algorithms." arXiv preprint arXiv:2005.03557 (2020)
>
> [R5] Chen, Tianyi, et al. "Communication-efficient policy gradient methods for distributed reinforcement learning." IEEE Transactions on Control of Network Systems 9.2 (2021): 917-929.

---

> > ### Author Response · Authors · 2022-11-19
> > **Response to Reviewer ZMya's Comments [Part 2]**
> >
> >
> >
> > > **Your Comment 2:** The algorithm's update is poorly explained. From my view, it's just SPIDER's update [1], which is a popular variance reduction approach. However, there is no discussion with existing variance reduction methods
> > [1] Spider: Near-Optimal Non-Convex Optimization via Stochastic Path Integrated Differential Estimator.
> >
> > **Our Response:** Thanks for your comments. It is true that our variance reduction (VR) method is inspired by SPIDER-type update. However, compared to using SPIDER to solve conventional minimization problems, it is the complex nature of the *nonconvex-strongly-concave min-max problem* of RL policy evaluation that makes the use and analysis of SPIDER-type algorithms highly challenging. Specifically, note that the policy evaluation problem is first reformulated as a nonconvex-strongly-concave optimization problem.
> > So far, how to design SPIDER-type algorithms for solving the resultant nonconvex-strongly-concave problem and its convergence performance analysis remain open in the literature. Specifically, in our primal-dual algorithmic design, the updates on the primal and dual variables need to be done simultaneously. Conducting theoretical analysis for algorithms with VR techniques on top of such primal-dual algorithmic framework is dramatically different and far more challenging than those of SPIDER-type algorithms in conventional minimization problems.
> >
> >
> >
> > At the time of submission, due to space limitation, we only reviewed the following related work in the main body of the paper: i) the paper related to TD Learning with function approximation for policy evaluation, and ii) the non-convex-strongly-concave minimax optimization paper. We do agree with the reviewer that it is important to also review related VR algorithms in nonconvex optimization. In this revision, we have added the literature review on VR algorithms for nonconvex optimization in the Appendix.
> >
> >
> > > **Your Comments 3:** The projection step in (8) is not explained.
> >
> > **Our Response:** Thanks for your question. The projection step in (8) is the most widely used projection operator, which is defined as follows:
> >
> > ${{\omega}}^{(k+1)}=\mathcal{P} _{\mathcal{W}}( {{\omega} }^{(k)}+\alpha  G _{{\omega} }^{(k)})\triangleq \underset{\widetilde{\omega} \in \Omega}{\operatorname{argmin}}\|\tilde{\omega}-({{\omega} }^{(k)}+\alpha G _{{\omega} }^{(k)})\|^2.$
> > We have added the above expressions in our new version of the paper. The projection operator helps us to ensure the $\omega \in \mathcal{W}$. $\Theta$ is a convex set and  $\Theta\subseteq \mathbb{R}^d$. We note that, in the Appendix of this revision, we have also added new convergence analysis for a more general problem with $\theta\in \Theta$ and $\omega \in \mathcal{W}$, where $\Theta\subseteq \mathbb{R}^d,\mathcal{W}\subseteq \mathbb{R}^d$ are convex sets.

---

> > > ### Author Response · Authors · 2022-11-19
> > > **Response to Reviewer ZMya's Comments [Part 3]**
> > >
> > >
> > > > **Your Comment 4:** There are too many typos and unprecise claims, making the paper hard to read. Here are examples:
> > > -The last $g_\theta(s)$ in (1) should not have a transpose.
> > > -The strongly-concavity definition in assumption $4.2$ is wrong; the function should also be additionally concave.
> > > -The $L_{-} f$-smoothness definition in assumption $4.3$ is unprecise; theta and omega should be in the same norm.
> > >
> > > **Our Response:**
> > >
> > > 1. Thanks for pointing out this typo. We have fixed this typo in our revised version. We would also note that this typo does not affect the methods and major conclusions in our paper.
> > >
> > > 2. It appears there is some misunderstanding here. There is actually no issue in the strong concavity definition in Assumption 4.2. Note that $\mathcal{L}(\theta,\omega)$ is differentiable in $\omega$. Following from the standard definition of strong concavity, we have: $\mathcal{L}(\theta,\omega) \leq \mathcal{L}(\theta,\omega') +\nabla _{\omega}\mathcal{L}(\theta,\omega')^{\top}(\omega-\omega')-\frac{\mu}{2}\|\omega-\omega'\|^2$, $\forall \boldsymbol{\omega}, \boldsymbol{\omega}' \in \mathbb{R}^{d}$, $\mu>0$ and fixed $\theta$. Although the previous expression is the standard definition of strong concavity, it is important to note that there are several *equivalent* conditions for strong concavity, such as the one we used in our paper, i.e., there exists a constant $\mu>0$ such that $\|\nabla _{\boldsymbol{\omega}}\boldsymbol{\mathcal{L}}(\boldsymbol{\theta} ,\boldsymbol{\omega} )-\nabla _{\boldsymbol{\omega}}\boldsymbol{\mathcal{L}}(\boldsymbol{\theta} ,\boldsymbol{\omega} ')\|\geq \mu \|\boldsymbol{\omega} -\boldsymbol{\omega}'\|$, $\forall \boldsymbol{\omega}, \boldsymbol{\omega}' \in \mathbb{R}^{d}$. This equivalent definition can be proved by using the monotone gradient condition and the Cauchy-Schwartz inequality. A proof of this fact can be found in Lemmas 2 and 3 in [R2].
> > > Due to the above equivalence, the function does not need to have an additional concave property.
> > >
> > > [R1]Zhou, Xingyu. "On the fenchel duality between strong convexity and lipschitz continuous gradient." arXiv preprint arXiv:1803.06573 (2018).
> > >
> > >
> > > 3. We are a bit confused by this comment. We guess the reviewer meant $L_f$-smoothnes in Assumption 4.2 rather than in 4.3. If this is the case, again, it appears there is some misunderstanding here. There is actually no issue in the $L_f$-smoothness definition in Assumption 4.2. In Assumption 4.2, we assume that both $\nabla _{\theta} \mathcal{L} _ {i} (\theta, \omega )$ and $\nabla _{\omega} \mathcal{L} _{i} ({\theta},{\omega})$ have $L_f$-smoothness property, thus we have $||\nabla _{\theta}{\mathcal{L}} _{i }({\theta} ,{\omega} ) -\nabla _{\theta}{\mathcal{L}} _{i}({\theta} ',{\omega} ')|| \leq   L_f \left( ||{\theta} -{\theta} '||+||{\omega} -{\omega} '|| \right)$
> > >
> > >  ;  $||\nabla_{\omega}{\mathcal{L}} _{i }({\theta} ,{\omega} ) -\nabla _{\omega}{\mathcal{L}} _{i}({\theta} ',{\omega} ')|| \leq   L _f \left( ||{\theta} -{\theta} '||+||{\omega} -{\omega} '||\right)$. Thus, the stated inequality in Assumption 4.2 follows from adding the previous two inequalities and the triangle inequality. In this revision, to avoid similar confusion in the future, we have rewritten the original inequality as two separate inequalities mentioned above.
> > > Also, we note that the *same* stated assumption has also been used in [Wai et al. (2019)], [Luo et al. (2020)], [Qiu et al. (2020)].
> > >
> > > We hope the above explanations could clarify the reviewer's doubts.

---

### Official Review · Reviewer_Mbzc · 2022-10-25

**Confidence:** 4
**Correctness:** 3
**Technical Novelty And Significance:** 2
**Empirical Novelty And Significance:** 3
**Recommendation:** 5

**Clarity, Quality, Novelty And Reproducibility:**

The paper is clearly written with words and sentences of good quality, and the single-timescale method for the given problem is novel in literature to my best knowledge. The complexity result matches the one in Luo et al in the regime $\kappa=O(1)$, and hence somewhat expected.

**Strength And Weaknesses:**

Page 2, middle part.
I am confused when the authors indicate "new proof techniques in convergence rate analysis". To quote words in your paper it "resolves an open question and clarifies an ambiguity in the state-of-the-art convergence analysis ...". Can you clarify what was the ambiguity and how did you clean it?

Page 3, middle part (related work).
My understanding is that Du et al. (2017) study the TD(0) formulation of policy evaluation with linear function approximation, which can be re-cast as a strongly-convex-concave problem. What is the true novelty of this work, in comparison with Du et al. (2017)? Du et al. (2017) solved this problem via VR-based methods and achieves a linear convergence rate via the gradient descent-ascent type method. Is it correct that nonlinear function approximation leads to possible non-concavity?

Page 4, first paragraph.
SREDA adopts "PiSARAH" subroutine but the latter is not detailed.

Page 4, second paragraph.
It should be desirable if the dependency on *all* problem-dependent constants should be considered when discussing matching the lower bounds. At first glance, it seems that the method is not sharp in terms of $\kappa$, and this is natural due to that no acceleration method is adopted. In my opinion, even if the resulting upper bound failed to match the lower bound by Zhang et al. (2021) in certain regimes, some discussions in fine-grained angles should be added.

Page 7, middle part (Remark after Theorem 1).
The author wrote "Theorem 1 states that VRPD achieves an $\mathcal{O}(1/K)$ convergence rate to an $\epsilon$-FOSP". The complexity and rate of convergence have been awkwardly mixed together and caused confusion. One should modify it accordingly. I like the more detailed stochastic sample complexity of $\mathcal{O}(M + \sqrt{M}\kappa\epsilon^{-1})$ where $\kappa$ denotes the condition number of the strongly convex part. Also, "The most challenging part in proving Theorem 1 stems from the fact that one needs to simultaneously evaluate the progress of the gradient descent in the primal domain and the gradient ascent in the dual domain of the minimax problem". This is merely a standard technique and is well-known in recent minimax optimization.

Some reference bibitems are duplicated, e.g. Sutton et al. Authors are encouraged to clean them when preparing their next version.

**Summary Of The Paper:**

The authors developed a new efficient, single-timescale variance-reduced primal-dual method (VRPD) with an emphasis on the feasibility of solving non-convex policy evaluation problem in reinforcement learning (RL) with nonlinear function approximation, also known as the value function estimation or on-policy learning with nonlinear function approximations. VRPD is a single-timescale algorithm that meets an $\mathcal{O}(1/K)$-rate of convergence to a first-order stationary point for general non-convex-strongly-concave (NCSC) minimax optimiztion, under mild assumptions and appropriate parameter choices. The authors claim that they provide the best-known result in the literature. An enhanced VRPD method is also proposed, namely VRPD${}^{+}$, to allow adaptive batch sizes using historical information while maintaining the theoretical convergence rate. The authors empirically validated the proposed method and concluded higher sample efficiency over comparable methods including VRPD.

**Summary Of The Review:**

In summary, the authors did a satisfactory job introducing the single-timescale method that solves an important practical problem (policy evaluation in RL), achieving sharp convergence rates matching the lower bound under certain regimes ($\kappa=O(1)$). With that said, the paper potentially admits some issues of overclaiming their results. Plus, the author seems to have solved a much broader problem but applied it to a very specific problem (of policy evaluation with nonlinear function approximation). Extending such a framework to wider application instances as the next step might be a good idea. As of the current form I give my rating of borderline inclining rejection and encourage the authors to address these main points in their revised version.

---

> ### Author Response · Authors · 2022-11-19
> **Response to Reviewer Mbzc's Comments [Part 1]**
>
> > **Your Comment 1:** Page 2, middle part. I am confused when the authors indicate "new proof techniques in convergence rate analysis". To quote words in your paper it "resolves an open question and clarifies an ambiguity in the state-of-the-art convergence analysis ...". Can you clarify what was the ambiguity and how did you clean it?
>
> **Our Response:** Thanks for the comments. The clarification of your question can be found in the second paragraph in Section 2.1. In this revision of our paper, we added a reference on Page 2 to help readers better understand our paper's contribution. Specifically, there is an ambiguity in the convergence analysis result of the state-of-the-art paper on MSPBE-baed RL policy evaluation [Wai et al. 2019], which is detailed below:
>
> * ***Ambiguity:*** Although the convergence theorem in [Wai et al. 2019] appears to have a $\frac{1}{K}$ factor ($K$ being the total number of iterations), their convergence rate bound is in the form of $\frac{F^{(K)}+\text{Constant1}}{K\cdot\text{Constant2}}$ (cf. Theorem 1, Eq. (26) in [Wai et al. 2019]), which still depends on $K$. Notably, the $F^{(K)}$ term in the denominator in Eq.~(26) inherently depends on the primal and dual values ${\theta}^{(K)}$ and ${\omega}^{(K)}$ in the $K$-th iteration, respectively. It is unclear whether ${\omega}^{(K)}$ can be bounded in [Wai et al. 2019], hence leading to an **ambiguity** in guaranteeing an $\mathcal{O}(1/K)$ convergence rate. Thus, whether an $\mathcal{O}(1/K)$ convergence rate is achievable in single-timescale policy evaluation with nonlinear function approximation and constant step-sizes remains an **open question** thus far.
>
> * ***How we clean it:*** We bound per-iteration change in $J(\theta)$ instead of the function $\mathcal{L}(\theta^{(k)},\omega^{(k)})$ as in [Wai et al,2019]. This helps us avoid the technical limitations of [Wai et al,2019] and successfully establish the $\mathcal{O}(1/K)$ convergence rate. Specifically, the convergence result in [Wai et al,2019] relies on $F^{(K)} = \mathbb{E} [\mathcal{L}(\theta^{(0)},\omega^{(0)})- \mathcal{L}(\theta^{(K)},\omega^{(K)})]$. However, since $\mathcal{L}(\theta^{(K)},\omega^{(K)})$ is a non-convex-strongly-concave function, we cannot ensure the upper bound of $F^{(K)}$. Thus, the convergence results in [Wai et al,2019] still depend on $K$ and lead to ambiguity to the $\mathcal{O}(1/K)$ convergence rate. In contrast, our paper bound per-iteration change in $J(\theta)$ instead. The convergence result thus relies on $J(\theta^{(K)})$. Since $J(\theta)=\mathcal L(\theta,\omega^*) = \max_{\omega\in\mathcal{W}}\mathcal L(\theta,\omega)$,  the constant  $\mathbb{E}[J(\theta^{(0)})]-\mathbb{E}[{J}(\theta^{(K)})]$ is always upper bounded by $\mathbb{E}[J(\theta^{(0)})]-\mathbb{E}[{J}(\theta^*)]$, where $\theta^*= \min_{\theta} J(\theta)$. This is how we clean the $K$-dependence in our theoretical analysis.
>
> * ***Other differences between our paper and [Wai et al, 2019]:*** i) Besides the above ambiguity, the best convergence results of [Wai et al. 2019] only hold when the step-size is $\mathcal{O}(\frac{1}{M})$, where $M$ is the size of the dataset. Such a small step-size is problematic for modern RL problems with a large state-action transition dataset. In contrast, our paper allows the constant-level step-size. (ii) In the Appendix of our paper, we provide the convergence result for a more general problem where the parameter $\theta\in \Theta$ and $\omega \in \mathcal{W}$. $\Theta$ and $\mathcal{W}$ are convex sets and $\Theta\subseteq \mathbb{R}^d,\mathcal{W}\subseteq \mathbb{R}^d$. However, [Wai et al. 2019] is a special case of our problem where $\theta\in \Theta$ and $\omega \in \mathbb{R}^d$.
>
>
> In summary, this paper's key contribution and novelty are that we resolve the above open question by proposing two new algorithms, achieving an  $\mathcal{O}(1/K)$ convergence rate.

---

> > ### Author Response · Authors · 2022-11-19
> > **Response to Reviewer Mbzc's Comments [Part 2]**
> >
> > > **Your Comment 2:** Page 3, middle part (related work). My understanding is that Du et al. (2017) study the TD(0) formulation of policy evaluation with linear function approximation, which can be re-cast as a strongly-convex-concave problem. What is the true novelty of this work, in comparison with Du et al. (2017)?
> >
> > **Our Response:** Compared to [Du et al. (2017)], our paper has the following key differences:
> >
> > 1. ***Methodology:*** [Du et al. (2017)] uses linear approximation for the value function $V_{\theta}(\cdot)$. Thus, it is not applicable for nonlinear models, (e.g., deep neural networks). In contrast, our paper uses nonlinear approximation for the value function $V_{\theta}(\cdot)$. For RL policy evaluation with nonlinear function approximation, the MSPBE-based approach can be formulated as a nonconvex-strongly-concave min-max optimization problem, which is far more challenging than the strongly-convex-concave problem under linear function approximation in [Du et al. (2017)].
> >
> > 2. ***Algorithm:*** [Du et al. (2017)] uses a SVRG-type variance reduced algorithm. In contrast, our paper uses the state-of-the-art recursive SPIDER-type variance-reduced algorithm. Specifically, the SVRG-type estimator in [Du et al. (2017)] is constructed based on the information of the initialization of the outer loop of the algorithm. In contrast, the SPIDER-type estimator utilizes fresher information and yields a more accurate estimation of the full gradient (see Eqs. (7a)-(7b)).
> >
> > 3. ***Problem:*** [Du et al. (2017)] didn't consider constrained cases of the parameters. Specifically, [Du et al. (2017)] considered the case with $\theta \subseteq \mathbb{R}^d$, $\omega\in \mathbb{R}^d$, which is a special case of the problem we aim to solve in Eq. (4), where $\theta \subseteq \mathbb{R}^d$, $\omega\in \mathcal{W}$. In addition, in our Appendix, we also discussed a more general problem with parameters $\theta\in \Theta$ and $\omega \in \mathcal{W}$, where sets $\Theta\subseteq \mathbb{R}^d,\mathcal{W}\subseteq \mathbb{R}^d$ are convex.
> >
> > 4. ***Experiments:*** As shown in Fig. 5 in our paper, with the same amount of parameters, nonlinear approximation always achieves a smaller mean square error than that of the linear approximation approach in [Du et al. (2017)].
> >
> > > **Your Comment 3:** Du et al. (2017) solved this problem via VR-based methods and achieves a linear convergence rate via the gradient descent-ascent type method. Is it correct that nonlinear function approximation leads to possible non-concavity?
> >
> > **Our Response:** Thanks for your question. Yes, the nonlinear function approximation results in the nonconvex minimization optimization problem on the primal side. In contrast, the linear function approximation in [Du et al. (2017)] leads to the strongly convex optimization problem on the primal side. Both problems in our paper and [Du et al. (2017)] on the dual side is a strongly-concave maximization problem. We would like to note that for any fixed $\theta$, the objective function $\mathcal{L}({\theta}, \omega)$ is strongly concave in $\omega$ since $D$ in Eq. (1) in our paper is positive definite. Similar conclusions can also be found in [Wai et al. 2019], a paper also studies the reformulated MSPBE in policy evaluation, where the nonlinear approximation leads to nonconvex on the primal side problem, and the dual side problem is always a strongly-concave problem.
> >
> > > **Your Comment 4:** Page 4, first paragraph. SREDA adopts "PiSARAH" subroutine but the latter is not detailed.
> >
> > **Our Response:** The PiSARAH in SREDA (see ALgorithm 3 and 6 in [Luo et al., 2020]) is a subroutine that aims to help the SREDA algorithm achieve the desired accuracy at the initialization step. The PiSARAH-step can be seen as an additional step to solve an inner concave maximization subproblem. Note that our algorithm does not need such a subroutine. Thus, the computational complexity of SREDA is higher than ours. Thanks for your suggestion. We have added the above details on "PiSARAH" in our revised version.
> >
> > [Luo et al., 2020] Luo Luo, Haishan Ye, Zhichao Huang, and Tong Zhang. Stochastic recursive gradient descent ascent for stochastic nonconvex-strongly-concave minimax problems. arXiv preprint arXiv:2001.03724, 2020

---

> > > ### Author Response · Authors · 2022-11-19
> > > **Response to Reviewer Mbzc's Comments [Part 3]**
> > >
> > >
> > > > **Your Comment 5:** Page 4, second paragraph. It should be desirable if the dependency on all problem-dependent constants should be considered when discussing matching the lower bounds. At first glance, it seems that the method is not sharp in terms of $\kappa$, and this is natural due to that no acceleration method is adopted. In my opinion, even if the resulting upper bound failed to match the lower bound by Zhang et al. (2021) in certain regimes, some discussions in fine-grained angles should be added.
> > >
> > > **Our Response:** Thanks for your questions. Our sample complexity result matches the lower bound of Zhang et al. (2021)[R2] in terms of the dependence on $\epsilon$ and sample size $M$. We would also like to note that although our sample complexity doesn't match the state-of-the-art result in [R2] in terms of $\kappa$, our proposed algorithms still has the following advantages over [R2] in the following aspects:
> > >
> > >  1. The algorithm in [R2] contains an inner min-max subproblem (cf. Line 6 of Algorithm 1 in [R2]). Solving such subproblems in the inner loop incurs high computational costs. In contrast, our algorithm does not have such an inner min-max subproblem. Thus, to a certain extent, the computational complexity of the algorithm in [R2] is higher than ours.
> > >
> > >  2. The number of parameters in [R2] are far more than ours ($\alpha_t,\tau,\rho, q$) and it is hard to tune them for good convergence performance. Specifically, the algorithm requires $\alpha_t=\frac{\mu^5}{504 L^5}$, which requires the knowledge of the condition number. By contrast, our algorithm only requires step-sizes $\alpha$ and $\beta$ to be sufficiently small, which is much easier to tune in practice.
> > >
> > > 3. As shown in Line 7 of Algorithm 1 in [R2], the proposed algorithm requires an additional step $z_{k+1}=y_k^t+\frac{\sqrt{q}-q}{\sqrt{q}+q}\left(y_k^t-y_{k-1}^t\right)$, which requires more memory cost than our methods.
> > >
> > > 4. In order to solve the inner minimax problem in Line 6 of Algorithm 1 in [R2], the authors suggested using algorithms such as SGDA/SAGA/SVRG, instead of proposing their own method. In contrast, our proposed algorithm can directly solve the minimax problem and lead to a promising convergence result.
> > >
> > >
> > > 5. It is unclear whether the method in [R2] can be extended to problems with constraints. In contrast, our method can solve problems with $\theta \in \Theta, \omega \in \mathcal{W}$.
> > >
> > > [R2] Siqi Zhang, Junchi Yang, Cristóbal Guzmán, Negar Kiyavash, and Niao He. The complexity of nonconvex-strongly-concave minimax optimization. arXiv preprint arXiv:2103.15888, 2021.
> > >
> > >
> > > > **Your Comment 6:** Page 7, middle part (Remark after Theorem 1). The author wrote "Theorem 1 states that VRPD achieves an  convergence rate $\mathcal{O}(1/K)$ to an $\epsilon$-FOSP". The complexity and rate of convergence have been awkwardly mixed together and caused confusion. One should modify it accordingly. I like the more detailed stochastic sample complexity $\mathcal{O}\left(M+\sqrt{M} \kappa \epsilon^{-1}\right)$ where  $\kappa$  denotes the condition number of the strongly convex part.
> > > >
> > > **Our Response:** Thanks for the suggestion. Due to the space limitation, we put the convergence and complexity results together so save space in the original submission of our paper. Following your suggestion, we have added a corollary for sample complexity results in this paper revision.

---

> > > > ### Author Response · Authors · 2022-11-19
> > > > **Response to Reviewer Mbzc's Comments [Part 4]**
> > > >
> > > >
> > > > > **Your Comment 7:**  Also, "The most challenging part in proving Theorem 1 stems from the fact that one needs to simultaneously evaluate the progress of the gradient descent in the primal domain and the gradient ascent in the dual domain of the minimax problem". This is merely a standard technique and is well-known in recent minimax optimization.
> > > >
> > > >
> > > > **Our Response:** Thanks for your comments. It is true that simultaneously evaluating primal descent and dual ascent for the min-max optimization problem is relatively standard algorithms such as GDA or SGDA. However, such analysis for variance-reduction (VR) methods for minimax problems remains under-explored. What we truly meant earlier is that the most challenging part is doing the theoretical analysis of our proposed VR-based algorithm for min-max optimization problems. Here, we would like to further elaborate the challenges in simultaneously evaluating the progresses of primal and dual variables in our algorithm.
> > > >
> > > > First, the difficulty stem from the min-max optimization problem iteself. Solving min-max problems even for achieving stationarity could be extremely challenging: (i) The non-convexity of the primal-side objective will prevent us from finding a global optima solution; (ii) It is difficult to guarantee both primal and dual variables to the stationary solution at the same time since they and may influence each other; (iii) The major difficulty is to propose the sample-effiecient algorithm and provide its theoretical guarantees. As we discussed in our paper, [Wai et al, 2019] cannot ensures the $\mathcal{O}(1/K)$ convergence rate. In this paper, we proposed a strong convergence metric (see Eq. (9)), including the first-order stationery for the primal variable and optimality for the dual variable. Our new convergence metric is stronger than those in existing papers [Luo et al., 2020; Zhang et al., 2021], and we managed to establish a convergence rate of $O(1/K)$ based on this new convergence metric.
> > > >
> > > >
> > > > Second, we use the SPIDER-type variance reduction technique with the goal to achieve an $O(1/K)$ convergence rate. VR techniques are more complex than conventional methods for min-max optimization problems. Although existing works [ Luo et al., 2020, Xu et al.;., 2020a] also adopted VR techniques in solving min-max optimization problems, these algorithms assume a subroutine that can solve a maximization subproblem for the dual variables in each iteration. This signficantly simplifies their proofs. Providing theoretical analysis for algorithms with VR techniques implemented on both primal and dual sides without the above subroutine assumption is far more involved. Please see our proof details of Lemma 3 and see how We successfully overcome these difficulties.
> > > >
> > > >
> > > > Third, our paper focuses on a constrained setting, where $\omega \in \mathcal{W}$, where $\mathcal{W}\subseteq \mathbb{R}^d$ is a convex set. This setting is more difficult to analyze compared to the unconstrained setting $\omega \in \mathbb{R}^d$ [Zhang et al., 2021; Xu et al., 2020a]. To ensure the feasibility of the constraint $\omega \in \mathcal{W}$, we update $\omega$ with projection ${\omega}^{(k+1)}=\mathcal{P} _{\mathcal{W}}\left({\omega}^{(k)}+\alpha G _{{\omega}}^{(k)}\right)$ instead of ${\omega}^{(k+1)}={\omega}^{(k)}+\alpha G _{{\omega}}^{(k)}$. Thus, in the theoretical analysis, we cannot use the simple equation $G _{{\omega}}^{(k)}=\frac{1}{\alpha }({\omega}^{(k+1)}-{\omega}^{(k)})$. Rather, we have to try another method to bound $({\omega}^{(k+1)}-{\omega}^{(k)})$. Specifically, we use the optimality condition of $\omega$ to overcome this issue. Please see proof details in Lemma 5.
> > > >
> > > >
> > > >
> > > >
> > > > [Luo et al., 2020] Luo Luo, Haishan Ye, Zhichao Huang, and Tong Zhang. Stochastic recursive gradient descent ascent for stochastic nonconvex-strongly-concave minimax problems. arXiv preprint arXiv:2001.03724, 2020.
> > > >
> > > > [Xu et al., 2020a] Tengyu Xu, Zhe Wang, Yingbin Liang, and H Vincent Poor. Enhanced first and zeroth order variancereduced algorithms for min-max optimization. 2020a.
> > > >
> > > > [Zhang et al., 2021] Siqi Zhang, Junchi Yang, Cristóbal Guzmán, Negar Kiyavash, and Niao He. The complexity of nonconvex-strongly-concave minimax optimization. arXiv preprint arXiv:2103.15888, 2021.
> > > >
> > > >
> > > >
> > > > > **Your Comment 8:** Some reference bibitems are duplicated, e.g. Sutton et al. Authors are encouraged to clean them when preparing their next version.
> > > >
> > > > **Our Response:** Thanks for catching these errors. We have fixed these errors in the revised version of our paper.

---

> > > > > ### Author Response · Authors · 2022-11-19
> > > > > **Response to Reviewer Mbzc's Comments [Part 5]**
> > > > >
> > > > >
> > > > > > **Your Comment 9:** In summary, the authors did a satisfactory job introducing the single-timescale method that solves an important practical problem (policy evaluation in RL), achieving sharp convergence rates matching the lower bound under certain regimes ($\kappa = O(1)$). With that said, the paper potentially admits some issues of overclaiming their results.
> > > > >
> > > > > **Our Response:** Thanks for the positive feedback. According to your comments and suggestions above, we have modified and clarified our theoretical claims in this revision of our paper.
> > > > >
> > > > > > **Your Comment 10:** Plus, the author seems to have solved a much broader problem but applied it to a very specific problem (of policy evaluation with nonlinear function approximation). Extending such a framework to wider application instances as the next step might be a good idea. As of the current form I give my rating of borderline inclining rejection and encourage the authors to address these main points in their revised version.
> > > > >
> > > > > **Our Response:** Thanks for your comments and suggestion. Yes, you are correct that our proposed method is applicable for solving general nonconvex-strongly-concave min-max problems. We also agree that it would be interesting to go beyond RL policy evaluation and apply our algorithm to a wider range of applications with min-max problem structure, which will surely increase the impact of our work in the optimization area. Although our work has other potential applications in optimization, we would like to emphasize that RL policy evaluation is our main motivation for this work and one of the most relevant applications to showcase the significance of achieving low sample complexities. Also, the exact min-max optimization formulation is the standard framework widely studied in the literature for RL policy evaluation (see, e.g., [Wai et al. 2019], [Qiu et al. 2020], and [Liu, et al. JAIR’18]).

---

### Official Review · Reviewer_vYTZ · 2022-10-26

**Confidence:** 3
**Correctness:** 3
**Technical Novelty And Significance:** 2
**Empirical Novelty And Significance:** 2
**Recommendation:** 6

**Clarity, Quality, Novelty And Reproducibility:**

The paper is clearly written and easy to follow. The proofs seem clear to me. The only comment I have is to define \epsilon_\theta in the proof of Theorem 1.

In terms of quality and novelty, the algorithm design is not new. A similar VR gradient design was used in Nonconvex smooth optimization in the past. Applying this to the extended nonconvex strongly concave minmax problem is new in the literature if I am not missing something. Although I am not updated with the state-of-the-art in this specific area.



**Strength And Weaknesses:**

The convergence results are provided for the first time and they look theoretically sound to me.

Weakness:
1. I am not sure about the role of \Theta in the paper. It seems that it is mentioned in the last paragraph on page 4 and then dropped subsequently in the min-max formulation (2) or (4). Would the analysis work if such \Theta is explicitly imposed on the min-max problem? Note that in such a case, the \|J(\theta)\| may not converge to zero.

2. The maximization over \omega \in R^d of equation (2) changes to \omega \in \mathcal{W} in equation (4). Does that change the policy evaluation problem? There is no justification provided for this abrupt change. Is compactness of \mathcal{W} necessary for convergence?

3. Why does the paper say it is a single timescale algorithm? I see two timescales \alpha and \beta in the algorithm. Are they set the same in the implementation?

4. I don't see how the sample complexity is O(\kappa/\eps) in Theorem 1. Based on the step-size policy, I believe it should be a larger power of \kappa. E.g., \kappa^3/\eps.

3. Since the main technical thrust of the paper is VR algorithms for the min-max problem, I was expecting some literature review of such algorithms, at least for nonconvex settings. It is important to put this result in the context of such literature and show the novelty of their work over the state-of-the-art in the optimization literature. I recommend that the authors include papers/articles working on VR algorithms for smooth nonconvex (note that J(\theta) is a smooth nonconvex function) and nonconvex-concave minx-max problems.



**Summary Of The Paper:**

This paper proposes two variance-reduced (VR) algorithms for a specific nonconvex strongly concave min-max problem obtained after reformulating the policy evaluation problem as a min-max game with nonlinear function approximation for the value function of the policy. They provide the first O(1/K) convergence and the possibility of removing the full gradient computation required in VR methods in certain iterations.

**Summary Of The Review:**

Overall, the paper makes some new contributions to both areas: VR algorithms for minmax problems and Policy evaluation in RL. Although, the paper does lack significantly in terms of literature review. It ignores the first area completely. The single-timescale part is fuzzy at best. The order of condition number in the sample complexity seems wrong to me.

---

> ### Author Response · Authors · 2022-11-19
> **Response to Reviewer vYTZ's Comments [Part 1]**
>
> > **Your Comment 1:**  I am not sure about the role of $\Theta$ in the paper. It seems that it is mentioned in the last paragraph on page 4 and then dropped subsequently in the min-max formulation (2) or (4). Would the analysis work if such $\Theta$ is explicitly imposed on the min-max problem? Note that in such a case, the $|J(\theta)|$ may not converge to zero.
>
>
> **Our Response:** Thanks for your question. In the last paragraph on Page 4, the notation $\theta \in \Theta \subseteq \mathbb{R}^d$ is meant for generality purposes, where $\Theta$ could be equal to $\mathbb{R}^d$. In the original submission of our paper, from the min-max formulation (2) onward, we relaxed $\Theta$ in the problem from $\theta \in \Theta$ to $\theta \in \mathbb{R}^d$ for simplicity. We agree with the reviewer that we should have clarified this in the paper. We have added this clarification in this revision.
>
> Moreover, we remark that it is not difficult to extend our algorithm on solving the problem with $\theta \in \Theta$. As shown in the appendix in the modified version, we can update the $\theta$ with the projection operator: ${{\theta} }^{(k+1)}=\mathcal{P} _{{\Theta}}({{\theta} }^{(k)}-\beta G _{{\theta} }^{(k)})$  instead of the original updating rule: ${{\theta} }^{(k+1)}={{\theta} }^{(k)}-\beta G _{{\theta} }^{(k)}.$ Consequently, we can ensure the updating parameter $\theta \in \Theta$ at each iteration $k$.
>
> As for the theoretical convergence analysis, the major difference between this revision with $\theta \in \Theta$ and the original submission is in Lemma 1. Specifically, we can no longer use the equation ${{\theta} }^{(k+1)}- {{\theta} }^{(k)} =-\beta G_{{\theta} }^{(k)}$ as in the original submission, but utilizing the property of the compact set of $\Theta$ to bound the ${{\theta} }^{(k+1)}- {{\theta} }^{(k)}$. See the proof details in the revised of proofs in this revision.
>
> Also, we agree that $|J(\theta)|$ may not converge to zero if the set $\Theta$ is not appropriately chosen. We note that the main focus of the theoretical part of our paper is on solving a more general nonconvex- strongly-concave minimax optimization problem with $\theta\in\Theta$. Besides, the subspace $\Theta$ can be as large as the $\Theta=\mathbb{R}^d$, thus ensuring $|J(\theta)|\rightarrow 0$. Due to the nonconvexity of the problem, our work aims to find a stationary (or saddle) point solution (not a global optimal solution) and the associated convergence rate of our proposed algorithm.
>
>
> > **Your Comment 2:** The maximization over $\omega\in R^d$ of equation (2) changes to $\omega\in \mathcal{W}$ in equation (4). Does that change the policy evaluation problem? There is no justification provided for this abrupt change. Is compactness of $\mathcal{W}$ necessary for convergence?
>
> **Our Response:** Thanks for your questions. First, we remark that, in policy evaluation problems, letting $\omega\in \mathcal{W}$ is unnecessary. In our original submitted version, we let $\omega\in \mathcal{W}$ for analytical simplicity, so we can use the Danskin Theorem to conclude that $\nabla J(\theta) = \nabla_{\theta} \mathcal{L}(\theta, \omega^*(\theta))$. In this revised version of our paper, we further provide Lemma 3 to prove that $\nabla J(\theta) = \nabla_{\theta} \mathcal{L}(\theta, \omega^*(\theta))$ without requiring the condition that $\omega\in \mathcal{W}$. Thanks for your comments, which definitely improve the clarity of this work.
>
> To summarize, TD learning with function approximation in Eq. (2), where $\theta \subseteq \mathbb{R}^d$, $\omega\in \mathbb{R}^d$, is a special case of the problem we aim to solve in Eq. (4), where $\theta \subseteq \mathbb{R}^d$, $\omega\in \mathcal{W}$.
>
>
>
> > **Your Comment 3:** Why does the paper say it is a single timescale algorithm? I see two timescales $\alpha $and $\beta$ in the algorithm. Are they set the same in the implementation?
>
> **Our Response:** In the min-max optimization literature (see, e.g., A.1 in [R1]), the definition for the single-timescale algorithm is as follows: the learning rates for primal variables and dual variables are of the same orders, i.e., there exist two constants $C>0$ and $C'>0$, such that $0<C<\alpha_k/ \beta_k<C'<\infty$ as iteration $k \rightarrow \infty$. Note that, in our paper, the learning rates $\alpha$ and $\beta$ are fixed constant step-sizes and $\alpha/\beta$ is also a fixed constant. Thus, our algorithms are single timescale algorithms following the aforementioned definition.
>
> [R1] Shuang Qiu, Zhuoran Yang, Xiaohan Wei, Jieping Ye, and Zhaoran Wang. Single-timescale stochastic nonconvex-concave optimization for smooth nonlinear td learning. arXiv preprint arXiv:2008.10103, 2020.

---

> > ### Author Response · Authors · 2022-11-19
> > **Response to Reviewer vYTZ's Comments [Part 2]**
> >
> > > **Your Comment 4:** I don't see how the sample complexity is $O(\kappa/\epsilon)$ in Theorem 1. Based on the step-size policy, I believe it should be a larger power of $\kappa$. E.g., $\kappa^3/\epsilon$.
> >
> > **Our Response:**  Thanks for pointing out this typo. The algorithm's sample complexity is indeed $\mathcal{O}(\kappa^3/\epsilon).$ We want to note that this doesn't affect the major conclusion of our paper, where our algorithm achieves an $\mathcal{O}(1/K)$ convergence rate.
> >
> >
> > > **Your Comment 5:** Since the main technical thrust of the paper is VR algorithms for the min-max problem, I was expecting some literature review of such algorithms, at least for nonconvex settings. It is important to put this result in the context of such literature and show the novelty of their work over the state-of-the-art in the optimization literature. I recommend that the authors include papers/articles working on VR algorithms for smooth nonconvex (note that $J(\theta)$ is a smooth nonconvex function) and nonconvex-concave minx-max problems.
> >
> > **Our Response:**  Thanks for your comments. In this paper, we focus on the optimization problem for the policy evaluation. We reformulate the policy evaluation problem as a non-convex-strongly-concave optimization problem, which is much more difficult than standard minimization problems. Thus, in the main paper, we only reviewed the paper related to the TD Learning with function approximation for policy evaluation and the non-convex-strongly-concave minimax optimization paper. However, we do agree that it is worth reviewing VR algorithms in nonconvex optimization problem to better put our work into comparative perspectives. However, due to space limitation, we have added the literature review on VR algorithms (SVRG, SAGA, SCGD, SPIDER, SpiderBoost) in nonconvex optimization in the Appendix and dicuss their relationships to the nonconvex-concave min-max problems.
> >
> >
> > > **Your Comments 6:** The paper is clearly written and easy to follow. The proofs seem clear to me. The only comment I have is to define $\epsilon_{\theta}$ in the proof of Theorem 1.
> >
> > **Our Response:** Thanks for pointing this out. We use the notations $\epsilon_{\theta}$ and $\epsilon_{\omega}$ for simplicity, where $\epsilon_{\theta} =\mathbb{E}||
> > 		\nabla_{\theta}{\mathcal{L}}(\theta^{((n_k-1)q)},\omega^{((n_k-1)q)})-G_{\omega}^{((n_k-1)q)}||^2$ and $\epsilon_{\omega}=\mathbb{E}||
> > 		\nabla_{\omega}{\mathcal{L}}(\theta^{((n_k-1)q)},\omega^{((n_k-1)q)})-G_{\omega}^{((n_k-1)q)}||^2$.
> > Since our variance reduction technique evaluates a full gradient for every $q$ iteration, we have $\epsilon_{\theta}=  \epsilon_{\omega}=0$. In this revision, we have deleted these potentially confusing parts and updated our proofs in the Appendix accordingly.
> >
> >
> >
> >
> > > **Your Comment 7:** In terms of quality and novelty, the algorithm design is not new. A similar VR gradient design was used in Nonconvex smooth optimization in the past. Applying this to the extended nonconvex strongly concave minmax problem is new in the literature if I am not missing something. Although I am not updated with the state-of-the-art in this specific area.
> >
> >
> > **Our Response:** Although variance reduction (VR) techniques have been widely adopted in conventional minimization optimization problems, the use of VR techniques in min-max optimization remains under-explored, particularly in the min-max MSPBE problem for reinforcement learning (RL) policy evaluation. Compared to the algorithms for solving conventional minimization problems, our algorithms are primal-dual algorithms, where we have two variables to be updated simultaneously. Thus, analyzing the convergence performance guarantees of our proposed algorithms is challenging. Also, our proposed algorithms achieve an $O(1/K)$ convergence rate for the min-max MSPBE optimization problem for RL policy evaluation with nonlinear function approximation, which is the best-known result in the literature so far. Our work also provides new insights for general min-max optimization problems and sheds new light on algorithms design in this area.

---

### Author Response · Authors · 2022-12-08
**Looking forward to receiving the feedback.**

Dear Reviewers:

We would like to thank you and all reviewers for the constructive and insightful comments!

We have made a sincere attempt to address all the reviewers' comments and have incorporated their suggestions. We hope the response and the revised draft address the reviewer's concerns and eagerly look forward to receiving their feedback.

Thank you,
Authors

---

### Decision · Program_Chairs · 2023-01-20

**Decision:**

Reject

**Justification For Why Not Higher Score:**

The novelty of the algorithm and its analysis is limited. The presentation is poor.

**Justification For Why Not Lower Score:**

N/A

**Metareview: Summary, Strengths And Weaknesses:**

This paper proposes a variance-reduced primal-dual method (VRPD) for policy evaluation, which can achieve a fast convergence speed with nonlinear function approximation.

Strengths:

It provides a detailed convergence analysis.

Weaknesses:

The novelty of the algorithm is limited.

The dependence on the condition number is not optimal.

The presentation is poor

The author's response partly addressed some concerns from reviewers. However, not all the reviewers are excited about this paper. Therefore, I recommend rejection.